# Fine-scale heterogeneity in population density predicts wave dynamics in dengue epidemics

Victoria Romeo-Aznar[1,2,3], Laís Picinini Freitas [4,5], Oswaldo Gonçalves Cruz[5], Aaron A. King [6,7,8] & Mercedes Pascual [1,8✉]

The spread of dengue and other arboviruses constitutes an expanding global health threat. The extensive heterogeneity in population distribution and potential complexity of movement in megacities of low and middle-income countries challenges predictive modeling, even as its importance to disease spread is clearer than ever. Using surveillance data at fine resolution following the emergence of the DENV4 dengue serotype in Rio de Janeiro, we document a pattern in the size of successive epidemics that is invariant to the scale of spatial aggregation. This pattern emerges from the combined effect of herd immunity and seasonal transmission, and is strongly driven by variation in population density at sub-kilometer scales. It is apparent only when the landscape is stratified by population density and not by spatial proximity as has been common practice. Models that exploit this emergent simplicity should afford improved predictions of the local size of successive epidemic waves.

[1] Department of Ecology and Evolution, University of Chicago, Chicago, IL, USA. [2] Departamento de Ecología, Genética y Evolución, and Instituto IEGEBA (CONICET-UBA), Facultad de Ciencias Exactas y Naturales, Universidad de Buenos Aires, Ciudad Universitaria, Pabellón 2, C1428EHA Buenos Aires, Argentina. [3] Mansueto Institute for Urban Innovation, The University of Chicago, Chicago, IL, USA. [4] Postgraduate Program of Epidemiology in Public Health - Escola Nacional de Saúde Pública Sergio Arouca - Fundação Oswaldo Cruz, Rio de Janeiro, Brazil. [5] Programa de Computação Científica - Fundação Oswaldo Cruz, Rio de Janeiro, Brazil. [6] Department of Ecology and Evolutionary Biology, University of Michigan, Ann Arbor, MI, USA. [7] Center for the Study of Complex Systems, University of Michigan, Ann Arbor, MI, USA. [8] The Santa Fe Institute, Santa Fe, NM, USA. ✉email: pascual@uchicago.edu

When a new pathogen emerges, how large will successive epidemic waves be? When the infections confer temporary or long-term immunity, the answer to this central question will depend on spatial scale in complex ways we do not yet sufficiently understand. In particular, the size of successive outbreaks will result from the interplay of herd immunity and transmission seasonality across a landscape determined by the distribution and behavior of the human hosts, which has been called the "spatiotemporal geometry of herd immunity"[1]. Outbreaks of seasonal influenza, for example, can differ from city to city along multiple axes: epidemic vs. endemic character, depth of inter-seasonal troughs, and duration and shape of epidemic waves[2–4]. City size can modulate the influence of climatic drivers of transmission in seasonal influenza[1], and differential crowding within cities of different size has been shown to affect the shape of COVID-19 outbreaks, with longer tails in larger populations[5]. Megacities continue to grow spatially in ways that encompass pronounced heterogeneity in population density and movement, yet the effects of the resulting fine-scale structure on disease spread remain largely unexplored[6–8], with some notable exceptions relying on individual-based models[9]. What patterns have been observed have been deduced from implicit treatments of average crowding and connectivity as functions of city size[1,5]. However, transmission is an intrinsically local process and the local density and structure of the population has the potential to be a critical determinant of infection spread. Therefore examining the role of fine-scale population structure on infectious-disease dynamics is essential to improve predictive models of urban disease transmission and spread[10,11].

Traditional mathematical models with 'well-mixed' transmission between individuals within a population have formed a foundation for predictive epidemiological theory[12,13]. Explicit consideration of the spatial dimension is proving increasingly important[8,10,14,15] due to growing population connectivity at regional to planetary scales, novel sources of data on fine-scale individual movement, and the pronounced heterogeneity of the distribution of the human population across the landscape[16,17]. Whereas connectivity among cities and regions has been addressed with metapopulation formulations that couple local dynamics via movement fluxes[15,18,19], the treatment of space within cities remains a challenge[20,21]. It remains unclear at what scales aggregation of data is appropriate and how best to manage the trade-off between model fidelity and computational expense in the parameterization of movement, local environmental conditions, and epidemiological dynamics.

The foregoing issues are prominent in the case of vector-borne arbovirus infections, including dengue, Zika, and chikungunya. The dengue virus, in particular, has become a global health threat affecting a large fraction of the world's population as it continues to expand its geographical range[22,23]. Because of their domesticated lifestyle and close association with human hosts, the mosquito vectors responsible for dengue transmission (and also Zika, chikungunya, and yellow fever), *Aedes aegypti* and *Aedes albopictus*, are also expanding their distribution under urbanization and climate change. The population dynamics of these vector-borne diseases exhibit nonlinearity (in part, a consequence of the immunity engendered by infection) and climate-driven transmission seasonality (caused by seasonal cycles in mosquito abundance) over the small spatial scales at which both hosts and vectors vary in density[24,25]. A recent well-mixed model for the city as a whole, applied to the emergence of DENV1 in the megacity of Rio de Janeiro, Brazil, illustrates the challenge, by failing to predict time to re-emergence[26]. Previous studies (e.g., refs. [27,28]) also indicate the importance of spatial structure to the population dynamics of the disease and the build-up of herd immunity in particular.

The high spatial resolution of dengue surveillance data from Rio de Janeiro (250 m by 250 m) provides an opportunity to address variation in human population density at a degree of granularity unprecedented for a whole city. During the five years from 2010 to 2014, Rio de Janeiro experienced three major dengue outbreaks, dominated first by the DENV1 serotype, then followed for two consecutive years by the emergent DENV4 serotype, then newly arrived in Brazil (Fig. 1A)[29–31]. The intermittent epidemic pattern of two to three peaks dominated by an emergent single serotype, is typical of dengue dynamics in many cities of South America[32,33]. We address here one prominent feature of these emergent epidemics, namely the ratio of consecutive peak sizes when a serotype first enters the city. This ratio varies widely across the city (Fig. 1C); we demonstrate that it does so as a function of highly localized population density. To understand this phenomenon, we examine the interaction of local herd immunity with seasonal transmission. Specifically, we show that sparsely populated areas experience short-lived outbreaks which reach herd immunity sooner than those in densely-populated areas. Seasonality plays a major role by interacting with the build-up of herd immunity. Because seasonal declines in mosquito abundance curtail the transmission season, dense areas are left with disproportionately more susceptible hosts at the end of the first wave. Accordingly, the relative size of successive waves is highly sensitive to the timing of the introduction of infection into each local area. We investigate this "spark rate" of infection importation empirically, and examine its dependence on population density (computed here as the population in an area of 250 m by 250 m). We go on to investigate alternative representations of space in predictive models (Fig. 1B). We propose that spatial geometry based on human density at fine scales is more relevant to disease dynamics than the traditional coupling based on proximity in regular grids or arbitrary administrative units. This suggestion carries implications for predictive metapopulation models, including those for infectious diseases other than vector-transmitting ones, such as seasonal influenza and COVID-19.

## Results

**Empirical pattern in the size of successive epidemic peaks**. We specifically analyze the peak ratio for the two consecutive years of DENV4 during which this serotype, unlike DENV1, was new to the city (Fig. 1A)[29,30]. We consider that the whole population was initially susceptible to the virus, which neglects heterotypic protection. We find that the ratio of the peak sizes for the second and first seasons of DENV4 varies across the city with values below and above one, and exhibits a clear nonlinear relationship with human density (Fig. 2A, C). The peak ratio is larger at high and low densities than it is at intermediate values (Fig. 2A). A similar pattern is obtained for the ratio of the accumulated incidence in each season (Supplementary Fig. 8). This pattern arises when units are aggregated by population density but disappears when aggregation is constrained by geographical contiguity as is typically done for administrative subdivisions (Fig. 2B) (see maps in Supplementary Fig. 7). We can expect differences, since the two criteria of aggregation generate a very different organization of the city (Fig. 1B). Notably, the pattern of peak ratio as a function of human density is invariant under the number of groupings considered, as illustrated by the different colors in Fig. 2A. Thus, this dependence becomes scale-independent over an order of magnitude in the number of groupings when aggregation is governed by population density itself.

**Role of population density: deterministic SIR model**. To explain the peak ratio pattern, we initially investigated the role of population density with a deterministic seasonal SIR model

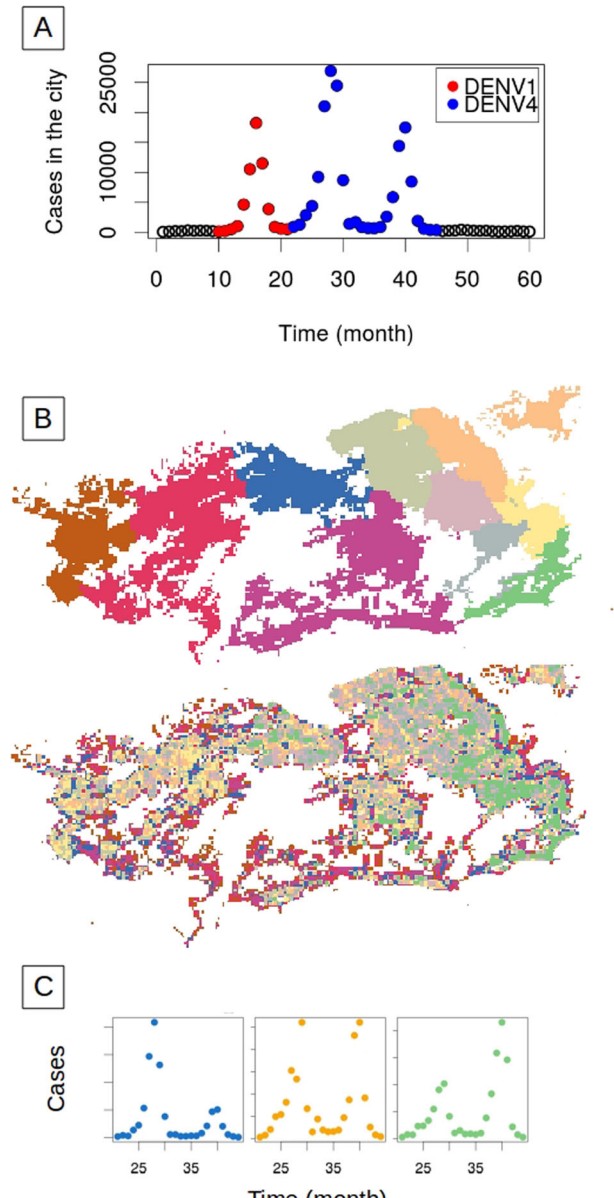

**Fig. 1 Dengue incidence patterns in Rio de Janeiro city. A** Total monthly cases of dengue reported in the city from January 2010 to December 2014. Red and blue dots correspond, respectively, to outbreaks with two different dominant serotypes, DENV1 and DENV4, with the latter making its first emergence in Rio de Janeiro. Black circles indicate seasons without outbreaks and only a small number of cases. **B** The maps illustrate the geography of the city when 250 m by 250 m units are aggregated into 10 strata by administrative regions (top) or population density—number of individuals in an area of 250 m by 250 m—(bottom) (see Supplementary Fig. 6 for the population density map with the scale reference). **C** Examples of three different possible patterns for the relative size of the peaks and therefore, peak ratio, for DENV4 incidence in three of the 10 administrative regions shown in the upper map in (**B**). The colors indicate the corresponding regions in this map.

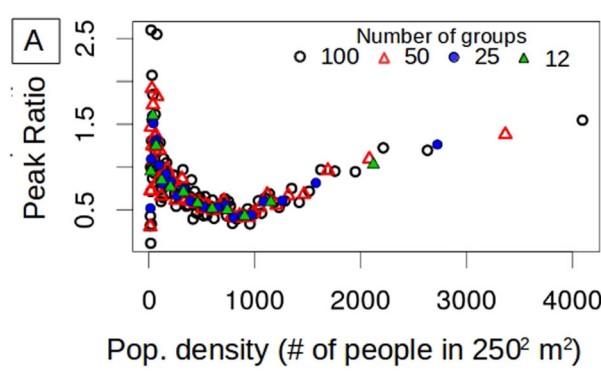

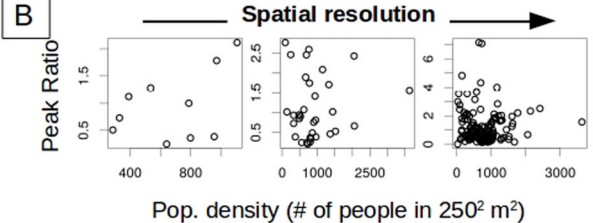

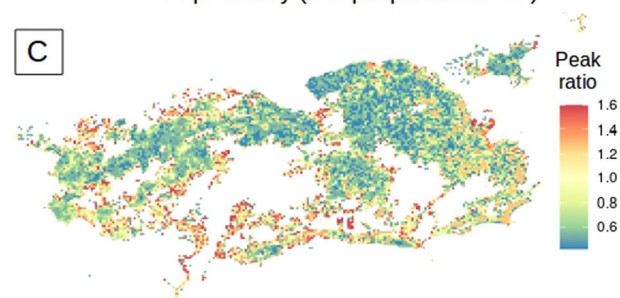

**Fig. 2 Ratio between the size of the successive peaks of DENV4.** The ratio for the size of the second epidemic over the first one was computed for each spatial location given different resolutions and the two different ways to aggregate space. In (**A**), the 250 m by 250 m units are aggregated according to their population density. In this case, the peak ratio exhibits a clear but nonlinear relationship with human density (see Supplementary Fig. 8D for a log scale representation). The colors correspond to partitions of the city into different numbers of groups. The pattern is invariant to the number of groups (resolution). In (**B**), space is subdivided according to the typical geographical partition into administrative units based on contiguous space. The city of Rio de Janeiro is administratively subdivided into different resolutions, namely 10 or 33 administrative regions (Fig. 1B and Fig. S7B, respectively) and 160 neighborhoods (Supplementary Fig. 7C). For any of these partitions, no relationship is observed between peak ratio and population density, as illustrated here for the three spatial scales of established administrative subdivisions of the city (from left to right: 10, 33, and 160 regions). In (**C**), the heterogeneity in peak ratio across the city is illustrated at the finest spatial resolution. The peak ratio spans a range of values, from below to above one (from blue to red), corresponding, respectively, to locations with a second peak smaller than the first one, and vice-versa. This fine-scale heterogeneity in peak ratio across space reflects that of population density (Supplementary Fig. 6).

(Methods). We hypothesize that two opposite variables shape the ratio of consecutive peaks by determining how much population immunity is accumulated during the first season, namely the arrival time of infection to a spatial unit and its population density. According to the model, given an arrival time $t_0$, the peak ratio increases with the population of the unit, with the second peak becoming larger than the first one (Fig. 3A). That is, smaller

units achieve the epidemic peak earlier, because their smaller susceptible pool is more rapidly depleted. When transmission rates vary seasonally, most of the susceptible population is depleted in the first year in a small unit, leaving few to be infected the next season. As the population density grows, the number of susceptible individuals remaining at the beginning of the second season is larger and the size of the second peak increases concomitantly. In addition to this effect, the timing of the local start of transmission also strongly affects the size of the pool of

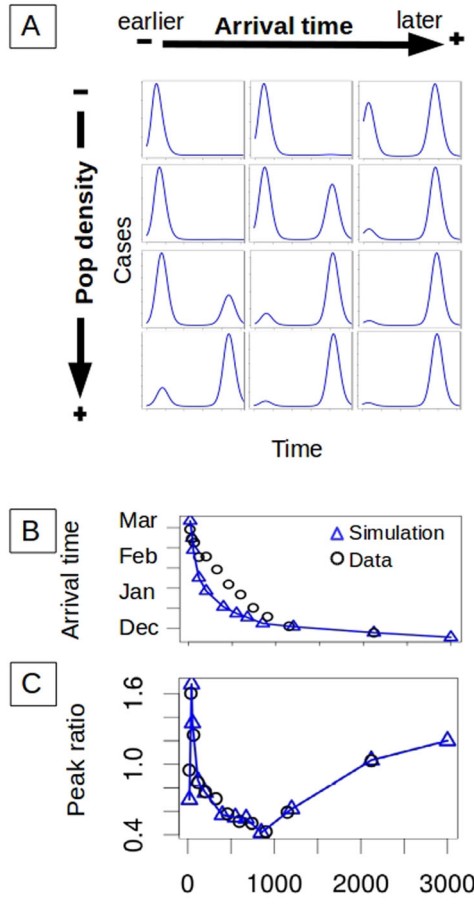

**Fig. 3 Deterministic SIR (susceptible–infected–recovered) dynamics and successive epidemic size. A** The temporal incidence of a unit is shaped by human density directly, but also indirectly via the arrival time of the first imported infection (the y-axis is normalized by the maximum value of cases for better visualization). The temporal dynamics are simulated with a deterministic SIR model with a seasonal transmission rate. For a given arrival time, the size of the second peak increases with the population density of the unit (top to bottom). The earlier the first infection reaches the unit, the smaller the size of the second peak (left to right). **B** Importantly, denser units are infected earlier (data: black circles). The blue triangles are used as input to the model to specify arrival time and therefore initial conditions in a given unit (with an exponential decay curve capturing the observed trend). The earlier importation of infection in denser units implies two opposite effects of population density on relative peak size (demonstrated in **A**). **C** For arrival times similar to those observed in the data, the model simulations (blue triangles) can capture the observed behavior (black circles) of the peak ratio with human density. The results correspond to a partition of the city into 12 groups according to population density.

susceptible individuals remaining after the first season. In particular, if infection arrives late, the local epidemic has less time to grow before the transmission season is curtailed. Thus, peak ratio increases with later arrival (Fig. 3A). We find that the time of local infection arrival is strongly associated with human density, whereby the most dense units exhibit the first reported DENV4 cases about three months earlier (Fig. 3B). Thus, population density affects peak ratio in two opposite directions, and the seasonal SIR model qualitatively recovers the documented nonlinear empirical pattern (Fig. 3C) when population densities and $t_0$ values comparable to those observed in Rio de Janeiro are used (Fig. 3B). The deterministic nature of the model combined with

the small size of the units makes simulations very sensitive to initial $t_0$ values. Small population sizes per se would introduce important demographic noise, here neglected, and the observed arrival times used in the simulations are likely delayed with respect to the first true local introduction of the virus. These limitations lead us to extend our analysis to a stochastic framework.

**Stochastic SIR model and spark rate**. To verify that our hypothesized effects are robust, we therefore consider a more realistic model that takes demographic stochasticity into account. To this end, we introduce the empirical rate of infection importation to a local unit $\sigma_u^{emp}$, referred hereafter as the "spark" rate, which allows us to sidestep the explicit coupling between the units. Without loss of generality, the spark rate and its estimation do not explicitly consider the source units from which infections are imported ("Methods"). A stochastic SIR model under well-mixed conditions applies within each unit, which allows for local extinction of infection and for the spark rate to re-initiate transmission. The initial conditions are self-contained in the model through the arrival of the first infection to a given unit. Specifically, for each unit $u$ the transmission rate is modeled as $\beta S_u I_u / N_u + \sigma_u$, where $\beta$ is the local transmission rate, $I_u$, $S_u$, and $N_u$ denote, respectively, the number of infected, susceptible, and total individuals in $u$, and $\sigma_u$ is the spark rate per unit. To take into account that we are working from observed cases, we consider a reporting rate $\rho \in (0, 1]$ and compute the spark rate as $\sigma_u = \text{Poisson}(\sigma_u^{emp}/\rho)$.

Armed with an estimated spark rate, we ask whether it can explain both the observed time of initiation of transmission at the unit level, and the pattern of peak ratio as a function of population density. We recover the observed delay in arrival time with population density, and find that this time is significantly affected by $\rho$ (Fig. 4A, see Supplementary Fig. 5 for the effects of other parameters). A small $\rho$ increases the spark rate, resulting in a tendency of earlier initiation of infection, but also decreases the detection of these early infections, which delays the observation of the first local case. Since detection of a single case is sufficient to determine arrival time, populated units are less affected by inefficient detection because they generate more local infections. The trade-off between these two effects of $\rho$ becomes increasingly unbalanced for larger population densities. Most importantly, the stochastic model predicts the empirical relationship of peak ratio with human density, and does so more accurately when it also better captures arrival times (Fig. 4B). The peak ratio is also affected by the initial levels of immunity in the population as Supplementary Fig. 14 shows. The ratio increases with the initial fraction of immune individuals, and this effect is stronger for medium and large population densities. However, the overall general pattern persists and the relationship with population density remains. In particular, an initial immune fraction of 10% produces a peak ratio pattern similar to that obtained when starting with the whole population susceptible to DENV4.

**Local and global determinants of the spark rate**. The stochastic model relied on an estimated spark rate. We now examine what factors determine this rate. We find a clear dependence on both a local and a global determinant, unit population density, and total city prevalence, respectively. A positive relationship with the total number of cases $C_{Tot}$ is expected, since more infection importations should be produced under higher levels of the virus circulating in the city. We find that the estimated spark rate grows as a power law with $C_{Tot}$ (Fig. 4C). This relationship is itself influenced by the local population density of the units, as illustrated with the different colors in Fig. 4C. More crowded areas would

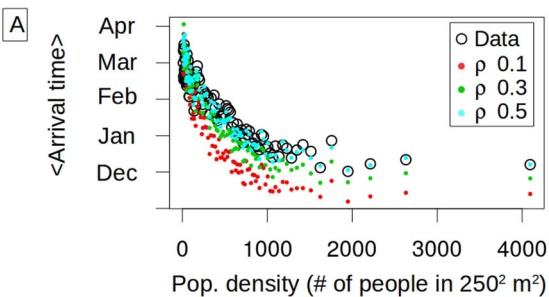

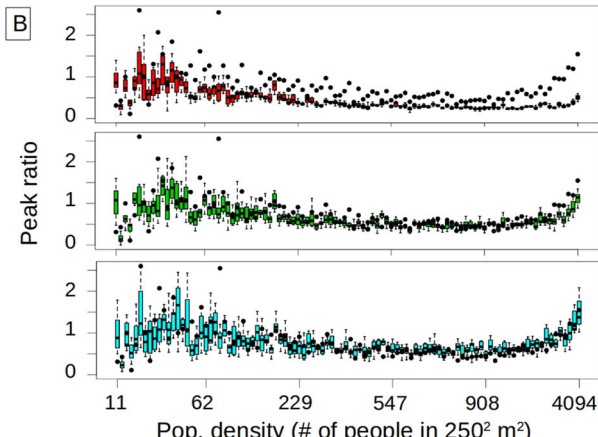

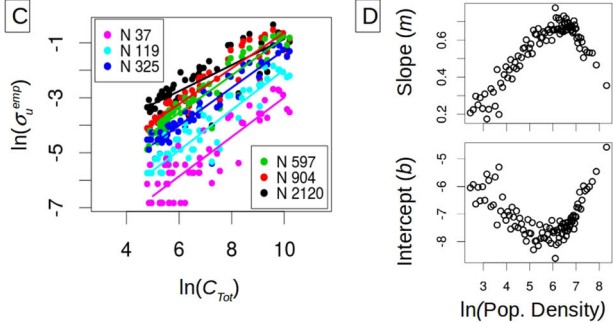

**Fig. 4 Stochastic simulations and empirical "spark" rate (number of sparks per month per unit). A** Mean observed arrival time as a function of human density. The original units are binned into 100 groups and arrival times are averaged for the units belonging to the same group. The arrival times computed from the data (black circles) are compared to those obtained in model simulations (with dots colored by reporting rate). **B** Peak ratio obtained in the simulations as a function of population density for the different reporting rates where the colors correspond to those in (**A**). The boxplots are computed from 20 stochastic realizations (in these, the box illustrates, as is standard, the median with the 25th and 75th percentiles, and the dotted lines indicate the extremes of the distribution). For comparison, the empirical values of peak ratio are also shown (in black, for the 100 groups). **C** Spark rate as a function of total incidence ($C_{Tot}$). The logarithm of the number of sparks (or number of imported infections) per month per unit exhibits a linear relationship with the logarithm of the total number of cases in the city. The more populated units receive a higher number of sparks as expected in a pattern that is well approximated by a power law. **D** The parameters of the power relationship between spark rate and total incidence vary as a function of population density. The different slopes ($m$) and intercept ($b$) from a linear regression to the log–log plot are shown in (**C**) as a function of the logarithm of population density. Thus, importation rate to a unit exhibits both a global and a local determinant, namely the total number of cases in the city and the local population in a given unit. These dependencies allow the specification of infection importation via a mean-field coupling and local conditions, circumventing the need to explicitly describe spatial connectivity at fine scales.

experience higher human movement fluxes than less dense ones, resulting in a higher probability of their inhabitants commuting to infected areas or receiving infected visitors. The spark rate increment with population size is nonlinear, increasing faster when densities are small, and saturating for the most populated units (Supplementary Fig. 4). To analyze these behaviors of the spark rate, we fit a linear relationship between the logarithm of the spark rate and $C_{Tot}$ as shown by the solid lines in Fig. 4C. The estimated parameters for each population group, the slope $m$ and intercept $b$, describe the influence of human density (Fig. 4D). These determinants of spark rate reinforce the important role of population density in the behavior of peak ratio. They also provide a handle to potentially reduce the complexity of the infection importation process.

## Discussion

Our results demonstrate that human density is a dominant driver of dengue dynamics at fine spatial scales comparable in size to city block and census tract. This effect scales up to explain the relative size of successive epidemic waves, a major epidemiological feature reflecting the interplay of seasonality with the depletion of susceptible individuals and the build-up of herd immunity. In other

words, this fundamental aspect of the dynamics of an immunizing infection is affected by variation in population density at fine spatial scales. Importantly, this does not mean that spatial aggregation or coarse graining of the landscape is not plausible. We find that the pattern of peak ratio with density is invariant to the number of spatial groupings over an order of magnitude, as long as coarser spatial partitions follow aggregation according to density itself, and not the traditional subdivision of administrative units based on typical contiguous space.

Thus, efforts to model dengue and possibly other infectious diseases in urban landscapes should consider the nature of aggregation space and not just its spatial resolution. On the one hand, administrative regions may better reflect similar environmental conditions such as temperature and socio-economic status influencing transmission intensity according to standard geography. On the other hand, the new partition we propose should by definition better capture human density and its effects on key aspects of dengue transmission such as infection spread and availability of susceptible individuals. Consideration of these different organizations of space can help identify and disentangle the effect of disease drivers, given that variation in incidence within the city occurs along both aggregation axes but for different sets of factors.

The effect of human density on peak ratio has practical relevance for informing public health efforts on the expected size of the next infection wave in different parts of a city. The documented peak ratio pattern demonstrates that the fine-scale spatial structure of urban populations strongly determines the temporal patterns of incidence at coarser resolutions. The importance of population structure was recently suggested by large-scale analyses of Covid-19 and influenza, in comparative studies of the temporal shape and endemicity of outbreaks at the whole-city level[1,5]. Here, we have explicitly described this structure through its effects for dengue.

The high-resolution dataset for Rio de Janeiro also revealed a clear dependence on human density of the seasonal timing of

infection arrival locally. This timing is critical to how much herd immunity will be acquired by the local population before the environmentally suitable transmission season ends, in the case of dengue in Rio de Janeiro due to variation in temperature and rainfall[34–36]. Crowded spatial units experience an earlier arrival date of the dengue virus, as previously reported for influenza[37]. The dependence of this timing on human density was successfully captured here by a stochastic model in which there is no explicit description of the spatial coupling between local units. Instead, the link between units is implicit in our model, via "sparks" arriving from unspecified locations from a global pool of city-wide infections.

Except for the spark rate, the parameters of the stochastic simulations were considered the same for all units and were not estimated from the case data. Based on previous estimates for dengue in the city, they are consistent with estimates of the reproductive number of the disease at this location and proved sufficient to explain the empirical patterns. An initial fraction of immune individuals below 10% is likely since the DENV4 strain responsible for the two outbreaks analyzed was not detected in the city before 2011[29–31]. Also, a reporting rate around 0.5 agrees with values documented for the city of Rio de Janeiro[38] and for other cities of Brazil, such as Porto Alegre[25] and Ceará (Fortaleza)[39]. In general, the proportion of clinically inapparent dengue in Brazil is estimated at around 40% (which implies a reporting rate of about 0.6), much lower than the world average of 75%[25]. Moreover, given that a total of 215,768 cases of dengue were reported during the analyzed outbreaks, a reporting rate below 0.4 would imply that more than 10% of the population was infected, a number that increases with the initial fraction of immune individuals (see Supplementary Fig. 13). For comparison, estimates based on serology indicated that by 2018 about 24% of the residents of the city had been exposed to dengue at some point in their lives[40], which makes it unlikely that half of these exposures would have happened only during the two studied years. In addition, small reporting rates values, as well as a large initial fraction of the population immune to the virus, are difficult to support if we compare the DENV cases produced during 2 years with the estimation of 16% of the population infected by COVID-19[41,42] (a much more transmissible virus) in about 36 months. Up to 10% protection would still be consistent with our results and could account for cross-protection from another serotype from previous seasons.

The patterns we have uncovered are for a city with intermittent seasonal epidemics. They raise the question of whether similar patterns arise in more endemic but still seasonal dengue regions. Exploratory analyses of a dengue dataset from Delhi suggests similar trends in peak ratio with population density (Supplementary Fig. 15). High-quality, high-resolution surveillance in such regions, including serotype assignment of the reported cases, would be valuable to examine the robustness of our findings across the geographical distribution of the disease and in the presence of co-circulating serotypes.

We have considered here frequency-dependent transmission because of its relevance to vector-transmitted diseases and many directly-transmitted ones. Future work should examine whether similar results hold for density-dependent transmission.

Although the spatial spread of infection involves the complex interplay of connectivity patterns and local transmission[25,43,44], our modeling of the spark process reveals that the effective result can be described in some systems in terms of two accessible quantities, namely the total number of cases in the city and the local human density. This finding suggests a novel formulation of metapopulation dynamics in urban environments that should be explored in future work, where space is aggregated according to population density and the coupling occurs through a global

incidence pool. Whether the complexity of human movement and resulting connectivity patterns can be captured in such a practical way in spatially explicit models of dengue and perhaps other infections remains an open question. This formulation combined with the sufficiently coarse partitions suggested by the scale-independent pattern we uncovered, provides an alternative to the intractable high-dimensional systems needed to resolve population density heterogeneity when modeling cities and geographical regions.

## Methods

### Data

*Spatial grid.* We created a grid whose units measure 250 m by 250 m based on the census tract layer for the city of Rio de Janeiro from the *Instituto Brasileiro de Geografia e Estatística* [Brazilian Institute of Geography and Statistics] (IBGE) website https://www.ibge.gov.br/geociencias/organizacao-do-territorio/malhas-territoriais. Uninhabited locations were excluded.

*Dengue cases on the grid.* Dengue is a disease of compulsory notification in Brazil, and cases are notified at the *Sistema de Informação de Agravos de Notificação* [Information System on Diseases of Compulsory Declaration] (SINAN). Dengue cases notified in Rio de Janeiro between January 2010 and March 2015 were geocoded according to address of residency, and then counted for each grid unit by the Secretariat of Health of the city. We obtained the monthly dengue cases data aggregated at the grid level.

*Population on the grid.* The population data is obtained from the Census 2010 (IBGE) (https://www.ibge.gov.br/estatisticas/downloads-estatisticas.html) and it is available at the census tract level. The census tract areas vary in size and can be bigger than the unit of the grid, primarily in the least densely populated zones of the city. To overcome this issue, we cropped from the census tract layer the areas classified as non-urbanized (such as water bodies, swamps, agricultural areas, green areas, beaches, rocky outcrops) in 2010 by the City Hall of Rio de Janeiro (layer available at http://www.data.rio/datasets/uso-do-solo-2010). The population of each census tract is distributed randomly (uniformly) in the areas obtained after deleting the non-urban areas. The population within the units is computed by adding the grid layer. To create the grid and edit the census tract layer we used QGIS (version 3.6.3)[45], and to obtain the population in the grid we used the R software[46] with the packages tidyverse[47] and sf[48]. We verify the accuracy of our estimated population by comparison with the WordPop dataset[49] (see detailed description and Supplementary Fig. 12 and Supplementary Note 2). We chose the WorldPop dataset because: (i) the estimates are also calculated based on census data and are available for 2010, (ii) the pixel size is 100 m, smaller than the size of our grid unit, and (iii) it is open access.

Since the units are in fact small and most of them conserve their area of 250 m by 250 m (Supplementary Fig. 1A), we consider population density as the population of each unit. For consistency, we do not consider units with small effective areas and/or populations sizes less than, or equal to, 10 in our analysis. In total, 8954/20212 units were so excluded. This choice circumvents the problem of high sensitivity to random population distribution, and urban vs. non-urban classification, in very small and/or sparsely populated areas. It also facilitates model simulation and does not affect the peak ratio pattern (Supplementary Fig. 1B).

### Peak ratio and spatial aggregation.

Since units are small, we binned them into $G$ groups and aggregated their times series of reported cases. The groups were generated according to two aspects: (1) the geographical location of the units as determined by the administrative divisions of the city (10 areas, 33 regions, and 160 neighborhoods); and (2) the population of the units based on quantiles in order to obtain equal size groups. We considered specifically four different partition levels, resulting in 12, 25, 50, and 100 groups with about 900, 450, 225, and 100 units, respectively (from a total number of 11,247 units for the whole city). Groups of unequal size can introduce different statistical effects (it is not the same, for example, to calculate a mean value using 1000 or 10 elements). To compare quantities across groups it is therefore prudent to define groups with the same number of elements. In particular, this consideration becomes important for a large number of groups. Since the population density distribution (number of individuals per unit) is not uniform, groups defined with "equidistant" boundaries would exhibit very different numbers of elements.

Given a unit $u$, we define its time series $\mathbf{v_u} = \{c_u(t_1), c_u(t_2), ..., c_u(t_f)\}$, where $c_u(t_i)$ is the number of reported cases of dengue at time $t_i$ ($i = 1, 2, ...f$) (and the bold symbol is used to indicate a vector). Thus, the aggregated time series is given by

$$\mathbf{V_g} = \sum_{u \in g} \mathbf{v_u} = \{C_g(t_1) = \sum_{u \in g} c_u(t_1), C_g(t_2) = \sum_{u \in g} c_u(t_2), ..., C_g(t_f) = \sum_{u \in g} c_u(t_f)\},$$

with $g = 1, 2, ..., G$. Then, for each $\mathbf{V_g}$ we computed the ratio between the sizes of

the second and first DENV4 peaks, that is

$$\text{peakratio}_g = \frac{max_{t \in season2}\{C_g(t_1), C_g(t_2), ..., C_g(t_f)\}}{max_{t \in season1}\{C_g(t_1), C_g(t_2), ..., C_g(t_f)\}} \quad (1)$$

(Supplementary Fig. 2).

**The deterministic SIR model.** Although dengue is a vector-borne disease, for simplicity we omitted the explicit representation of the dynamics of the mosquito population, and treated vector transmission via the seasonality of the transmission rate[26]. Thus, for each unit $u$, the deterministic SIR model is based on the following traditional differential equations:

$$\frac{dS_u}{dt} = \mu N_u - \beta S_u \frac{I_u}{N_u} - \mu S_u$$

$$\frac{dI_u}{dt} = \beta S_u \frac{I_u}{N_u} - \gamma I_u - \mu I_u \quad (2)$$

$$\frac{dR_u}{dt} = \gamma I_u - \mu R_u,$$

where $S_u, I_u, R_u$, are, respectively, the number of susceptible, infected, and recovered individuals, and $N_u$ the number of inhabitants, of the spatial unit $u$. Parameter $\mu$ is the mortality rate (equal to the birth rate), and $\gamma$ is the recovery rate. The seasonal transmission rate is specified as $\beta(t) = \beta_0(1 + \delta \sin(\omega t + \phi))$. The units are considered independent of each other, and the initial conditions establish that the whole population of each unit is susceptible to the virus ($S_u(t=0) = N_u$ and $I_u(t=0) = R_u(t=0) = 0 \forall u$). Transmission begins with one infected individual at a time $t_{0u} \geq t = 0$ where $t_{0u}$ is obtained from the data.

Since the goal of this model is to examine the representative dynamics of different population densities, we binned the units according to their population into 12 groups, and computed the mean value of their number of inhabitants $N_g = \langle N_{u \in g} \rangle$ and of their arrival times of the infection $t_{0g} \sim \langle t_{0u \in g} \rangle$ (where $g = 1, ..., 12$). We then simulated the system considering the 12 sets $\{N_g, t_{0g}\}$ as given.

**The stochastic model.** Since units will suffer local extinction of transmission, a major component of a stochastic implementation is the description of the local reintroduction of the virus, namely the arrival of a 'spark' or imported infection, in analogy to fire spread. Because space is described by a highly-resolved lattice, we considered that well-mixed transmission applies within each unit. Moreover, in lieu of explicit spatial coupling between units, we postulated the importation of infection through the specification of a spark rate.

For this purpose, we constructed a binary representation of the time series of cases per month by defining the spatial units either as positive or negative according to whether they reported cases or not (Supplementary Fig. 3). Then, to derive a spark rate we explored the dynamics of the number of positive units as follows,

$$U^+(t + dt) = U^+(t) + U_{new}^+(t, t+dt) - U_{extinct}^+(t, t+dt) \quad (3)$$

The number of positive units at time $t + dt$ is equal to the number of positive units at time $t$, plus the number of units that have been infected $U_{new}^+(t, t+dt)$ between $t$ and $t + dt$, minus the number of units that were infected at $t$ but are no longer infected at $t + dt$ (i.e., the number of 'extinctions' between $t$ and $t + dt$, $U_{extinct}^+(t, t+dt)$).

Since uninfected units (i.e., negative units) require the arrival of a spark to become positive, the following equation specifies the mean of $U_{new}^+(t, t+dt)$ under the assumption that a small unit is unlikely to receive more than a single spark in a period of time $dt$

$$\langle U_{new}^+(t, t+dt) \rangle \simeq N_{sparks}(t, t+dt)\frac{U^-(t)}{U}, \quad (4)$$

where $N_{sparks}(t, t+dt)$ is the number of sparks produced between $t$ and $t + dt$, $U^-(t)$ is the number of negative units at a time $t$, and $U$ is the total number of units in the city ($U = U^+ + U^-$).

By introducing Eq. (4) into Eq. (3) we obtain,

$$U^+(t + dt) \simeq U^+(t) + N_{sparks}(t, t+dt)\frac{U^-(t)}{U} - U_{extinct}^+(t, t+dt) \quad (5)$$

From Eq. (5) we can now compute the spark rate per unit $\sigma_u^{emp}(t, t+dt)$ from the high-resolution incidence data as

$$\sigma_u^{emp}(t, t+dt) = \frac{N_{sparks}(t, t+dt)}{U} \simeq \frac{U^+(t+dt) - U^+(t) + U_{extinct}^+(t, t+dt)}{U^-(t)} \quad (6)$$

In order to address the effects of human density on the spark rate, we binned the spatial units according to their population into $G$ groups. To avoid statistical effects due to group size, we considered population quantiles. Then, by applying Eq. (6) to each of these groups, we obtained an empirical spark rate per unit that depends on human density,

$$\sigma_{u \in g}^{emp}(t, t+dt) = \sigma_u^{emp}(t, t+dt; N_g), \quad (7)$$

where $N_g = \langle N_{u \in g} \rangle$ with $g = 1, 2, ..., G$.

*Simulations.* The associated differential equations of the stochastic model are those shown on Eq. (2) but the transmission component has now an additional term $\sigma_u$ to describe the importation of infections.

$$\frac{dS_u}{dt} = \mu N_u - \left(\beta S_u \frac{I_u}{N_u} + \sigma_u\right) - \mu S_u$$

$$\frac{dI_u}{dt} = \left(\beta S_u \frac{I_u}{N_u} + \sigma_u\right) - \gamma I_u - \mu I_u \quad (8)$$

$$\frac{dR_u}{dt} = \gamma I_u - \mu R_u$$

Since the inferred spark rate from the data (Eq. (7)) is obtained from observed infections, we computed the spark rate $\sigma_u$ as:

$$\sigma_{u \in g} = \text{Poisson}(\sigma_{u \in g}^{emp}/\rho) \quad (9)$$

where $\rho$ is the reporting rate.

The model shown on Eq. (8) was formulated as stochastic by incorporating demographic noise (with the different events represented as Poisson processes). It was implemented in R with the package **pomp**[50]. We also considered measurement error by assuming that the observed number of cases $C_u^{obs}$ during a period of time $T$ is,

$$C_u^{obs}(T) = \text{binomial}(\rho, C_u(T)), \quad (10)$$

where $C_u(T)$ is the number of cases computed in the unit $u$. We simulated the 11,247 units that compose the city of Rio de Janeiro, and aggregated the resulting time series as for the empirical data (see Peak ratio section).

The parameters of the model are given in Supplementary Table 1. We relied on parameters estimated for dengue transmission in Rio de Janeiro by ref. [26]. Those estimates were obtained for the aggregated city and for the emergence of DENV1. We use these parameters here as a sufficiently realistic set for illustrating and exploring the behavior of the stochastic model with population density. Moreover, with the exception of the spark rate, the model parameters were considered the same for all units. In particular, we applied a uniform reporting rate because access to the nearest public healthcare clinic does not show a dependency on population density (see Supplementary Note 1).

**Reporting summary.** Further information on research design is available in the Nature Research Reporting Summary linked to this article.

## Data availability

The data for the population and the time series of presence and absence of infections in each unit, as well as the aggregated time series of cases by administrative region and population density group, are available at[51]: https://github.com/vromeoaznar/DengueRio_peakRatio. Requests concerning the epidemiological raw data should be made to the Secretariat of Health of Rio de Janeiro city.

## Code availability

The code to produce the figures and to simulate the model is also available at[51]: https://github.com/vromeoaznar/DengueRio_peakRatio.

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

## Acknowledgements

The authors would like to thank Claudia T. Codeço for her comments on an earlier version of this work, and for her input on dengue in Rio de Janeiro. M.P. and A.A.K. would like to acknowledge the support of a collaborative grant from the National Science Foundation's Division of Mathematical Sciences and the U.S. National Institutes of Health (no. 1761612: Collaborative Research: Urban Vector-Borne Disease Transmission Demands Advances in Spatiotemporal Statistical Inference). A.A.K. was additionally supported by grants #1R01AI143852 and #1U54GM111274 by the U.S. National Institutes of Health. This research was completed with computer resources provided by the University of Chicago's Research Computing Center.

## Author contributions

V.R.-A. and M.P. conceived the study. V.R.-A. conducted the analyses. L.P.F. and O.C. conducted the preparation of the spatiotemporal data and geographic analyses for this purpose. All authors contributed to the interpretation of results. V.R.-A., M.P., and A.A.K. drafted the manuscript. All authors contributed to its final text.

## Competing interests

The authors declare no competing interests.
