## [Peer Review File · Nature Communications]

Fine-scale heterogeneity in population density predicts wave dynamics in dengue epidemicsREVIEWER COMMENTS

Reviewer #1 (Remarks to the Author):

This is a creative analysis of spatial and temporal dynamics of dengue infection in Rio de Janeiro. The exploration of dynamics in units aggregated by population size/density is a clean, straightforward analysis with really interesting results. The models that the paper proposes and uses to generate similar results are straightforward and reasonable. The results are relevant to many infectious pathogens and thus will be of general interest to the infectious disease dynamics community with general interest as well to theoretical ecology. I suspect a similar treatment of data from other infections may yield interesting insights in many cases. I have a few comments below that I hope will contribute to the manuscript.

It would be great if the authors considered alternative transmission terms, specifically incorporating a term that used other than a strict frequency dependence. My intuition would be that even slight density dependence could alter the relationship between peak ratios and population density as it counterbalances the extra time that it takes to exhaust susceptible populations in large population areas, but it would be interesting to hear the authors consideration of this.

One of my more moderate concerns about the robustness of results is the dependence of simulations to reporting rates. These rhos appear fairly large for what I would expect to be reporting rates for dengue and are quite large compared to what these authors estimated for an earlier period in their reference [25]. Case numbers appear to be roughly similar to that earlier period (with the population growing substantially). Is there any support for rho being as high even as 0.1 (rather than the more likely 0.02 reported in their earlier work)? The non-linear relationship between population density and peak ratios appears to be much more difficult to detect at these lower rhos. Can the authors comment on this?

Could the authors consider the possibility that rho varies spatially? What would be the effect of rho varying by population size?

How robust are the results to using ratios of cumulative incidence in each season rather than choosing peaks in each season? Given the mechanism, it seems like cumulative incidence would be the more robust measure to observe the relationship investigated here.

Is there evidence to support the fact that DENV4 had not emerged in Brazil before this time period? DENV4 had circulated ~20-30 years before. Wouldn't this contribute some level of immunity in the population? Are results dependent on the assumption that the population was completely susceptible to DENV4?

Why were the three regions shown in Fig. 1C chosen?

A suggestion, Figure 2A would be improved with a log x-axis in order to see the 0-200 range of population density more clearly.

Given the importance to the results, it would be good to compare the estimates of population arrived at by the processing decisions that the authors made in estimating the population in different spatial units with other sources of this data. Are there other sources of gridded populations available for this location?

Data availability

As I understand the statement of data availability, it would be impossible to reproduce the work presented here. I would encourage the authors to make the dengue surveillance data available at sufficient spatial scale to reproduce some of the analyses in the paper. These are aggregated and at least 6 years old and so their sensitivity appears to be low. Perhaps the authors can arrive at some level of aggregation to sufficiently deidentify data to allay privacy concerns but also make

some analyses reproducible.

Reviewer #2 (Remarks to the Author):

Review of NCOMMS-21-18516

Romeo-Aznar and colleagues present an intriguing hypothesis to explain the relative sizes of successive peaks in dengue epidemics within subunits of a large city. By introducing a metapopulation modeling approach that aggregates by population density rather than spatial proximity, the authors present evidence for density-dependent susceptible depletion and importation rates interacting with seasonally-forced transmission at an intra-city scale.

I enjoyed reading this manuscript. There are some potential issues with the analysis that I think need to be addressed before the conclusions are supported.

I will defer to other reviewers on dengue biology, for instance the assumption of neglecting heterotypic protection.

MAJOR COMMENTS

1. Spark rates and Reporting rate.

a. As I understand it the analysis is done conditional on a reporting rate which is assumed to be constant across spatial units. The reporting parameter is not estimated from data but rather the analysis is run at a range of different fixed reporting rates. It would be good to be more explicit about this.

b. How would it affect the analysis if reporting rate varied with population density? For example, what if reporting rate was systematically lower in denser areas of the city? Would that lead to higher estimates of the spark rate (σ_u) there? I would be curious to see the authors simulate this possibility to see if it offered an alternative explanation for the data.

c. Estimates of spark rate involve counting the number of spatial units that do not have reported cases. I would think for a given prevalence p in a subpopulation of size N , the probability of getting exactly 0 reports during a unit of time is proportional to $(1-p)^N$, which suggests that spark rate estimates could vary with prevalence and population size. This sampling effect would be operating in addition to the differential impacts of demographic stochasticity in small populations. Can the authors reject the null hypothesis that the scaling relationships they describe are due to this effect?

2. Writing / structure

The introduction develops the general importance of heterogeneity / structure in epidemic dynamics, but we get into the results without being clear what specifically the manuscript is going to test or how. Here are some suggested changes that may help

-consider moving the first paragraph of results at the end of the introduction.

-some phrases could be made less dramatic and more precise:

-"daunting heterogeneity": but what is daunting about the heterogeneity has not yet been explained when this phrase is used.

- "vast tracts of ground": is cinematic, yet has low information content: More precision would be better: by what measure are they large and larger than what?

- similarly, "traditional coupling of spatially proximal localities on Euclidean grids"  "aggregating based on spatial proximity"

MINOR COMMENTS

please include line numbers and page numbers in the future

in main text: "ratio [of consecutive peak sizes] varies widely across the city" to Fig legend 1c: I would mention explicitly that the three panes show three different possibilities for peak ratios

+ WE HYPOTHESE THAT "two opposite variables shape the ratio of consecutive peaks"

population size is sometimes used when I think the authors always mean population density: eg when referring to figure 3. As size and density can have different impacts I think it is important to be consistent/precise, and consider defining and providing units for population density - is the denominator per 250x250m unit, or per square km or?

Methods: Peak Ratio and Spatial Aggregation, should it be: "(10 areas, 33 regions and 160 neighborhoods)"?

I don't see U- defined before it appears in eq 3. Is it the number of negative / uninfected units? Similarly, I don't see an explicit definition of what N_sparks is.

around eqn 5, "to avoid statistical effects due to group size, we considered population quantiles." Which "statistical effects" are you referring to, and how does using quantiles avoid them?

spark modeling: I found the methods through equations 2-4 hard to follow. In eqn 3 are you assuming something similar to a Levins metapopulation model in ecology where the number of new occupied patches is given by the recolonization rate multiplied by the fraction unoccupied patches? Similarly, "combining equations 2 and 3" to get 4 - smaller more explicit steps would be helpful

Reviewer #3 (Remarks to the Author):

In their paper Victoria Romeo-Aznar and collaborators are interested in the role of space in the propagation of a disease, the influence of both the heterogeneity in population distribution and the population movement. They offer a relevant and quite elegant approach to the complex topic the spatial dimension in epidemiology. For their demonstration they have used Dengue as an example and datasets from Rio de Janeiro. Specifically the authors analyzed the role of spatial scale of data aggregation in the propagation of a pathogen in relation to human density. With a new space aggregation based on population density they proposed that human density at fine spatial scales could explain the relative size of successive epidemic waves.

I agree with the authors that transmission is first a local process and the local population structure is a key determinant of the pathogen propagation. And clearly this point is rarely adequately considered in the literature about Dengue. Thus one of the main advances of this manuscript is to highlight the importance of human density especially when human density is considered at a fine-

scale. The demonstration by the authors concerning this point is very convincing for Dengue propagation in the Rio de Janeiro city.

Another very interesting novelty proposed by the authors is to describe the importation of cases similarly to the arrival of a "spark". This process appears important to describe the transmission at a local level in absence of complete information about population movement. As stressed by the authors, this finding could suggest a novel formulation of metapopulation dynamics in complex urban environments.

The authors also found an invariant power law for the ratio of consecutive peak sizes when a serotype first enters in a given setting in relation to human density. This last point seems less convincing for several reasons. First the hypothesis of a naïve population to a pathogen, DENV4 in this manuscript, that first enters in a given setting appears very restrictive. Indeed as most tropical and sub-tropical regions have submitted the four dengue strains the potential applications of these results appear very limited. This greatly limits the significance of the presented results. I think that authors must explore the possibility of relaxing this hypothesis for instance by considering re-emerging dengue strains with a low seroprevalence (10 or 20%).

Secondly, there is just one example with two consecutive peaks... As DENV4 has emerged in different Brazilian cities, a second example, even with a coarser scale data, would reinforce this second main result.

Thirdly, the use of the parameter "arrival time of infection" could be problematic because it is a parameter very difficult to estimate in real world, especially for a disease dominated by asymptomatic cases and in crowded settings of under-developing countries. The sensitivity to uncertainty associated with this parameter could be done.

Fourthly, it is difficult to claim that the model found similar results to the observed ones when a key parameter of the model the "arrival time of infection" is obtained based on the same observations, observations that are used for the comparison with the model results (Fig 3B and page 6).

Finally, the importance of the reporting rate: It seems that the best adequation between the stochastic model results and the observations is for a reporting rate equals to 0.5 (Fig. 4).

However for a disease dominated by asymptomatic infections and in crowded and poor settings such as those we can find in Rio de Janeiro, the reporting rate should be very low... At least lower than 0.5...

For my point of view the authors did not give enough weight to the importance of the seasonality, particularly in the Discussion. In the case of Dengue, seasonality has the crucial role of stopping the transmission and then the depletion of human susceptibles,

Moreover the references about dengue spatial modeling quoted in the Introduction, could be completed by some others references, which seem to me important. Especially, Lourenço & Recker (2013) and Amaku et al (2016) that offer solution in a spirit quite similar with those of the authors. It is also the case for the reference of the SIR model used for dengue with just with a recent auto-reference [25], on a topic with a very expansive range of literature.

To summarize, the findings of this work are of potential interest in the context of spatial epidemiology. Nevertheless due to the aforementioned limitations, this manuscript is not ready, in my opinion, for publication. Once the imperfections have been removed, this manuscript, which presents a new and potentially interesting approach, could be published in a journal with a large audience such as Nature Communications.

Other comments

Abstract: "Models that exploit this emergent simplicity should afford improved predictions of epidemic waves."

This point is not demonstrated in this manuscript... Except the prediction of the size of the second peak that is very restrictive considering the potential complexity of an epidemic wave.

Same remark page 3.

Page 5: Regarding the neglected heterotypic protection: Is it reasonable hypothesis? Reference? This is an important point of discussion in multi-strains Dengue models. In your framework, what would the effect of heterotypic protection be?

Page 11: Are the presented results sensitive to the non-urban classification too?

Page 13: $t_u \geq t_0$: t_0 the "arrival time of infection"?

Page 13: "we specified no explicit spatial coupling between units": Reference?

Pages 13-15; Specify the difference between U and u, or perhaps it's just a font problem?

Figure 1: A scale for population density must be added.

Figure 3A: Is the same scale for the y-axis for the 12 graphs?

Figure 3: It is important to specify that the arrival times used in simulations are those obtained from the data.

References

- Lourenço J, Recker M (2013) Natural, Persistent Oscillations in a Spatial Multi-Strain Disease System with Application to Dengue. PLoS Comput Biol 9(10): e1003308.
- Amaku, M., Azevedo, F., Burattini, M. N., Coelho, G. E., Coutinho, F. A. B., Greenhalgh, D., ... & Massad, E. (2016). Magnitude and frequency variations of vector-borne infection outbreaks using the Ross–Macdonald model: explaining and predicting outbreaks of dengue fever. *Epidemiology & Infection*, 144(16), 3435-3450.

Response to Referees: Romeo-Aznar *et al* 2021

1. **Reviewer #1 (Remarks to the Author):** This is a creative analysis of spatial and temporal dynamics of dengue infection in Rio de Janeiro. The exploration of dynamics in units aggregated by population size/density is a clean, straightforward analysis with really interesting results. The models that the paper proposes and uses to generate similar results are straightforward and reasonable. The results are relevant to many infectious pathogens and thus will be of general interest to the infectious disease dynamics community with general interest as well to theoretical ecology. I suspect a similar treatment of data from other infections may yield interesting insights in many cases. I have a few comments below that I hope will contribute to the manuscript.

We thank the referee for her/his appreciation of the work and for the insightful comments.

- 1.1. It would be great if the authors considered alternative transmission terms, specifically incorporating a term that used other than a strict frequency dependence. My intuition would be that even slight density dependence could alter the relationship between peak ratios and population density as it counterbalances the extra time that it takes to exhaust susceptible populations in large population areas, but it would be interesting to hear the authors consideration of this.

We focused on a frequency-dependent term because it applies to vector-transmitted diseases, as well as to many important directly-transmitted infections such as measles, COVID-19, influenza, etc.. In particular, models for the population dynamics of dengue are always frequency-dependent, including in explicit coupled mosquito-human models. We have now made this choice clearer. We nevertheless included this interesting point as an open question in the Discussion (lines 280-282), as we do not have ourselves a clear expectation for the density-dependent case, and its numerical exploration would strongly depend on model parameters. Their values affect the time to susceptible depletion and therefore the peak ratio pattern, in addition to the specific formulation of the transmission term. We specifically used parameters from the literature for dengue in Rio de Janeiro, and these apply to the frequency-dependent model.

- 1.2. One of my more moderate concerns about the robustness of results is the dependence of simulations to reporting rates. These rhos appear fairly large for what I would expect to be reporting rates for dengue and are quite large compared to what these authors estimated for an earlier period in their reference [25]. Case numbers appear to be roughly similar to that earlier period (with the population growing substantially). Is there any support for rho being as high even as 0.1 (rather than the more likely 0.02 reported in their earlier work)? The non-linear relationship between

population density and peak ratios appears to be much more difficult to detect at these lower rhos. Can the authors comment on this?

Thank you for this important comment as we had not provided adequate explanation/discussion of the reporting rate values. This key parameter can change in time depending on several factors such as level of awareness and knowledge about the virus by the population and sanitary institutions, government budget allocated for DENV surveillance, etc. Importantly, the article in reference [25] analyzes case data from 1986 to 1988 in Rio de Janeiro when dengue first emerged in the city after decades without any reported cases. Therefore, neither the population nor the health institutions and the local government were prepared to detect and report infections efficiently, at least not at the level they have been in the last decade with a well-established surveillance system (including a recent now-casting system).

In the two years analyzed in our article, the city of Rio de Janeiro reported a total of 215768 cases. A simple calculation shows that a reporting rate of 0.02 (similar to the value we estimated for the emergence of DENV1 in 1986) would imply about 10.8 million total cases for the two years, which exceeds the whole population of the city - ~ 6 million. (This calculation considered that the whole population was susceptible when DENV4 arrived; if we relax this assumption and consider fewer initial susceptible individuals, we obtain an even higher number of infections (see Fig 1 below). A reporting rate of 10% would imply about 36% of the population infected in two years. This represents a fairly large and unrealistic value: for comparison, the 2018 prevalence estimated from serology for the city, for all DENV serotypes, is of ~ 20%¹. In addition, 36% of the population infected is quite large when compared with the current COVID-19 epidemic. In about one year and a half, if we consider for example a reporting rate of 40%², the prevalence of this more transmissible virus is about 0.16 (there have been ~397000 reported cases of COVID-19 by 07/29/2021³). By comparison, a reporting rate of 50% would imply 431536 cases of DENV4 in two years, which represents 7.2% of the population.

¹ Périssé, A. R. S., Souza-Santos, R., Duarte, R., Santos, F., de Andrade, C. R., Rodrigues, N. C. P., ... & Sobral, A. (2020). Zika, dengue and chikungunya population prevalence in Rio de Janeiro city, Brazil, and the importance of seroprevalence studies to estimate the real number of infected individuals. *Plos one*, 15(12), e0243239.

² Paixão, B., Baroni, L., Pedroso, M., Salles, R., Escobar, L., de Sousa, C., ... & Ogasawara, E. (2021). Estimation of COVID-19 under-reporting in the Brazilian States through SARI. *New Generation Computing*, 1-23.

³ <https://www.fast-trackcities.org/data-visualization/rio-de-janeiro-covid>

By conducting a transmission chain reconstruction of dengue in the city of Porto Alegre (2013-2016), Guzzetta et al (Nat Comm, 2018)⁴ obtained a reporting rate equal to, or higher than, 0.5. They also reported that “In Brazil, the proportion of clinically inapparent dengue is consistently estimated at around 40%, much lower than the world average of 75%.”, which translates to a reporting rate of about 0.6. In line with these values, the serology study of Rodrigues et al⁵ estimated a reporting rate of 40% for Ceara (Fortaleza), and Busch et al⁶ obtained a reporting rate between 0.25 and 0.75 for the city of Rio de Janeiro (for the 2012 dengue outbreak).

We therefore think that the range of values we considered for the reporting rate are justified, and that the literature supports an increasing value of this parameter from 1986 to 2012, with a well-established surveillance system in the city as the result of dengue becoming an emergent regional public health concern during this time. We have provided further explanation of this choice and of the contrast with our earlier estimate (in a new paragraph in the Discussion section, lines 249 to 270)

⁴ Guzzetta, G., Marques-Toledo, C. A., Rosà, R., Teixeira, M., & Merler, S. (2018). Quantifying the spatial spread of dengue in a non-endemic Brazilian metropolis via transmission chain reconstruction. *Nature communications*, 9(1), 1-8.

⁵ Rodrigues, E. M., Dal-Fabbro, A. L., Salomao, R., Ferreira, I. B., Rocco, I. M., & Fonseca, B. A. (2002). Epidemiology of dengue infection in Ribeirão Preto, SP, Brazil. *Revista de saude publica*, 36(2), 160-165.

⁶ Busch, M. P., Sabino, E. C., Brambilla, D., Lopes, M. E., Capuani, L., Chowdhury, D., ... & Glynn, S. A. (2016). Duration of dengue viremia in blood donors and relationships between donor viremia, infection incidence and clinical case reports during a large epidemic. *The Journal of infectious diseases*, 214(1), 49-54.

Fig 1. Percent of the total population infected in two seasons as a function of the initial fraction of the population immune at t_0 , for different values of the reporting rate (indicated to the right of the plot with different symbols and colors). The bottom graph zooms in into the smallest range of initial fraction immune.

1.3. Could the authors consider the possibility that rho varies spatially? What would be the effect of rho varying by population size?

We could not tell for sure whether the reviewer comment is about the effect of the reporting rate on the empirical or on the simulated pattern. We nevertheless find it very interesting and related to comments by the other referees. Given the lack of information about a spatial reporting rate at high resolution and in the absence of evidence for its spatial change, studies have typically considered reporting rates that are uniform in space, including large-scale studies (where units range from cities to countries). To address this concern here, we took two approaches described below: the first one assumes a hypothetical

scenario of how reporting rate changes with population density; the second presents empirical evidence on the access to reporting clinics throughout the city to support our choice of a constant rate.

We first analyzed the peak ratio pattern from the data with different dependencies of the reporting rate on population density. We can assume that $C_u^{real} \sim C_u^{obs} / \rho_u$, where $\rho_u = f(N = N_u)$ (N is population size and u denotes the unit). In particular, we analyzed a linear reporting rate that decreases with population (we chose a decreasing behavior to be in line with the comment of referee #3; also because we lack information on the specific form of this function, a simple linear relation is a sensible choice). Given that the studied quantity is the ratio of peaks sizes, and both peaks are equally affected by the reporting rate, the empirical pattern is not affected by a spatial reporting rate (see Fig 2A, below).

We also ran stochastic simulations with the linear function for the reporting rate with population. Fig 2,B2 shows that the resulting peak ratio pattern (and arrival time Fig 2,B1), for a reporting rate that varies between 0.3 and 0.5, is similar to the one obtained with a constant reporting rate of 0.5. The better accuracy of the pattern compared to the data with this linear relationship than with the constant reporting rate of 0.3 for the denser units (compare to Fig 4A and B in the main text) is because here, the denser population quantiles are wide. That is, the denser population group includes population densities that go from about 3000 to 8000 people per unit and therefore a reporting rate that varies from about 0.44 to 0.3.

We also considered whether there is evidence to assume that the reporting rate should vary significantly with population density for the city of Rio de Janeiro, where dengue transmission has been a major concern for public health authorities. We find that there are sufficient health units covering the city so that most people have access to one relatively close to their home, free of charge (see Fig 3A below). All these public clinics have to report dengue cases, as this is mandatory. Thus dengue notification includes all the people that seek assistance and does not result from a sample. That is, surveillance is passive. A bias could emerge though from people seeking assistance from private clinics, a behavior that would apply to a small fraction of the population concentrated in the richer neighborhoods. Because rich neighborhoods in Rio can be very dense (e.g. Copacabana), and as much as poor

communities (e.g. slums), we do not expect this sample bias to affect our results.

We specifically analyzed the access to healthcare clinics as a key factor affecting passive surveillance as implemented in Rio. We considered the distribution of the distances to the nearest public healthcare unit by population density group, shown in Fig 3B. As population density increases, there is a decreasing trend on the mean distance to the nearest public healthcare clinic, that goes from 1.6 km to 0.6 km. Despite the trend, the difference in mean distances is small. A maximum increase of 1km in distance should not impact the demand for medical assistance. We have highlighted the assumption of a uniform reporting rate in the Stochastic Model subsection (lines 456 to 459), and have added the above analysis (with more details than those presented here) in the Supplemental Material (title: Healthcare clinics' distance, FigS9-S11).

Fig 3. A) Map of the city showing the location of the public healthcare clinics (yellow dots) and the population density (population per unit). B) Box plot of the unit distance to the nearest public healthcare clinic as a function of unit population.

- 1.4. How robust are the results to using ratios of cumulative incidence in each season rather than choosing peaks in each season? Given the mechanism, it seems like cumulative incidence would be the more robust measure to observe the relationship investigated here.

Thank you for this comment. We now computed the ratio of cumulative incidence in each season and found that it shows the same pattern as the peak ratio (see Fig 4 below). Of course the range of the ratio changes some, as expected, but both quantities exhibit the same order of magnitude and the same pattern. We have added the plots below to the revised Supplemental Material Section (Fig S8), and a comment about this in the results section (line 119-120). Moreover we computed the ratio of cumulative incidence for a dengue dataset from Delhi (Fig S15) and added a paragraph about this in the main text (lines 272 to 278)

- 1.5. Is there evidence to support the fact that DENV4 had not emerged in Brazil before this time period? DENV4 had circulated ~20-30 years before. Wouldn't this contribute some level of immunity in the population? Are results dependent on the assumption that the population was completely susceptible to DENV4?

The reviewer is right about DENV4 circulation in Brazil ~ 30 years ago. “The first report of DENV4 genotype II (DENV4-II) in Brazil was in 1981–1982 in Boavista, the capital city of the northernmost state of Roraima. After a limited outbreak there, DENV4 was absent from the country until 2005–2007, when it was detected in the northern Amazon state that neighbours Roraima. In 2010 DENV4 was found again in other northern states”⁷ (the new strains of genotype II were genetically different from those isolated in the 1980s⁸). However, Brazil is geographically extensive and DENV4 was not detected in the state of Rio de Janeiro until 2011⁹. The genomic study by Faria et al. reported that the first human case of DENV4-II in the city of Rio de Janeiro occurred around Nov-Dec 2011.

A similar geographical spread was observed for CHIKV. This emergent virus was first detected in 2014 in Amapa (whose distance to Rio is similar to that of Roraima), and two years later, in Rio de Janeiro¹⁰.

In addition to the published information above, we ran the stochastic model with 10% of the population immune to the virus. That is, we considered the following initial conditions: $R_u(t=0)=0.1*N_u$, $I(t=0)=0$ and $S_u(t=0)=N_u-I_u(t=0)-R_u(t=0)$ (see Fig. 5 below). Since the virus was not detected in the city before 2011 and recently Périssé et al.¹¹ have reported a DENV-seroprevalence (for all serotypes) of about 24% for the city of Rio de Janeiro for 2018, a 10% level of immunized people represents a fairly conservative upper bound scenario for DENV4 transmission in 2011. For comparison, we also ran the model with 50% of the population immune to the virus as the initial condition. For the set of parameters shown on Table S1, we can see that the peak ratio increases as the initial fraction of the immune population increases, and that this effect appears stronger for medium and large population

⁷ Faria, N. R., Da Costa, A. C., Lourenço, J., Loureiro, P., Lopes, M. E., Ribeiro, R., ... & Sabino, E. C. (2017). Genomic and epidemiological characterisation of a dengue virus outbreak among blood donors in Brazil. *Scientific reports*, 7(1), 1-12.

⁸ Heringer, M., Souza, T. M. A., Monique da Rocha, Q. L., Nunes, P. C. G., da C Faria, N. R., de Bruycker-Nogueira, F., ... & Dos Santos, F. B. (2017). Dengue type 4 in Rio de Janeiro, Brazil: case characterization following its introduction in an endemic region. *BMC infectious diseases*, 17(1), 1-9.

⁹ Buonora, S. N., Passos, S. R. L., do Carmo, C. N., Quintela, F. M., de Oliveira, D. N. R., dos Santos, F. B., ... & Daumas, R. P. (2015). Accuracy of clinical criteria and an immunochromatographic strip test for dengue diagnosis in a DENV-4 epidemic. *BMC infectious diseases*, 16(1), 1-9.

¹⁰ De Souza, T. M. A., Ribeiro, E. D. A., Damasco, P. V., Santos, C. C., Bruycker-Nogueira, D., Chouin-Carneiro, T., ... & Dos Santos, F. B. (2018). Following in the footsteps of the chikungunya virus in Brazil: the first autochthonous cases in Amapá in 2014 and its emergence in Rio de Janeiro during 2016. *Viruses*, 10(11), 623.

¹¹ Périssé, A. R. S., Souza-Santos, R., Duarte, R., Santos, F., de Andrade, C. R., Rodrigues, N. C. P., ... & Sobral, A. (2020). Zika, dengue and chikungunya population prevalence in Rio de Janeiro city, Brazil, and the importance of seroprevalence studies to estimate the real number of infected individuals. *Plos one*, 15(12), e0243239.

densities. However, the overall general pattern persists and the relationship with population density remains. See in particular that a 10% initial immune fraction produces a peak ratio pattern quite similar to that obtained when starting with the whole population susceptible to DENV4.

We have now added these results to the manuscript (see Fig S14, Results section lines 180-186, and Discussion section lines 249-270, in main text).

1.6. Why were the three regions shown in Fig. 1C chosen?

They were chosen to illustrate different possibilities for the peak ratio of successive seasons that would be seen when incidence is aggregated according to the 10 administrative regions. This is now clarified in the caption of Fig 1.

- 1.7. **A suggestion, Figure 2A would be improved with a log x-axis in order to see the 0-200 range of population density more clearly.**

We have added a log-version of Fig 2A in the supporting material (see caption of Fig 2 and Fig S8D)

- 1.8. **Given the importance to the results, it would be good to compare the estimates of population arrived at by the processing decisions that the authors made in estimating the population in different spatial units with other sources of this data. Are there other sources of gridded populations available for this location?**

Thank you for this valuable suggestion. We have identified the following sources of gridded populations:

Dataset	Source	Grid Cell Size	2010 data availability	Source for National Level Population Totals	Distribution Policy
Gridded Population of the World (GPW), version 4	Center for International Earth Science Information Network (CIESIN) - Columbia University	~1 km	Yes	1) official country totals from census, and 2) Country totals adjusted to United Nations Population Division (UNPD) estimates and projections	Open access
Global Human Settlement Layer – Population (GHS-POP)	European Commission Joint Research Centre (JRC) and Center for International Earth Science Information Network (CIESIN) - Columbia University	~250 m, ~1 km	No	United Nations Population Division (UNPD) estimates and projections	Open access
Global Rural Urban Mapping Project (GRUMP)	Center for International Earth Science Information Network (CIESIN) - Columbia University; International Food Policy Research Institute (IFPRI), The World Bank, Centro Internacional	~1 km	No	United Nations Population Division (UNPD) estimates and projections	Open access

	de Agricultural Tropical (CIAT)				
History Database of the Global Environment (HYDE) Population Grids, version 3.1	Netherlands Environmental Assessment Agency (PBL)	~10 km	No	Population estimates are generally on the high end of the range of past estimates	Open access
LandScan Global Population database	Oak Ridge National Laboratory (ORNL)	~1 km	Yes	US Census Bureau	Commercial / Free for research use
World Population Estimate	ESRI	150 m (2016), 250 m (earlier)	No	Country-official estimates with 134 countries processed further by Michael Bauer Research GmbH.	Commercial / Free to ArcGIS Users
WorldPop	WorldPop	~100 m	Yes	1) Country-official estimates, and 2) United Nations Population Division (UNPD) estimates and projections	Open access

Source: Popgrid Data Collaborative. Global Population Grids: Summary Characteristics. Available at <https://www.popgrid.org/data-docs-table1>.

We chose to compare our estimates with the WorldPop dataset because i) the estimates are also calculated based on Census data and are available for 2010, ii) the pixel size is of 100m, smaller than the size of our grid unit, and iii) it is open access.

Briefly, the WorldPop¹² dataset contains the estimated total number of people per grid-cell at a resolution of 3 arc (approximately 100m at the equator). The mapping approach is Random Forest-based dasymmetric redistribution¹³.

¹² Worldpop, WorldPop :: DOI: 10.5258/SOTON/WP00645, (available at <https://www.worldpop.org/doi/10.5258/SOTON/WP00645>).

¹³ F. R. Stevens, A. E. Gaughan, C. Linard, A. J. Tatem, Disaggregating census data for population mapping using random forests with remotely-sensed and ancillary data. *PLoS One*. **10**, e0107042 (2015).

The total Rio de Janeiro's population was 6307639 according to the WorldPop dataset, and 6316636 in our grid. According to the National Census, there were 6320446 inhabitants in Rio de Janeiro city in 2010. Therefore, our total population is closer to the official one.

To compare the estimates spatially, for each unit i of our grid (measuring 250x250m), with $i = 1, 2, \dots, 20212$, we calculated the population counts of the overlapping 100m pixels from the WorldPop dataset.

Please find below the plot depicting the correlation between our estimates and the WorldPop's one for Rio de Janeiro city, 2010:

The populations are highly correlated ($R^2=0.88$). The dashed line represents a 1 to 1 relationship, and this line is very close to the regression one (the solid line).

We also performed a visual inspection of both sources of gridded populations. We observed that the WorldPop allocated population to uninhabited areas like swamps, forests, and other types of green areas. As explained in the Methods section, we cropped from the census tract layer the areas classified as non-urbanized (such as water bodies, swamps, agricultural areas, green areas, beaches, rocky outcrops) in 2010 by the City Hall of Rio de Janeiro (layer available at <http://www.data.rio/datasets/uso-do-solo-2010>). As a result, the

population is shown to be more accurately distributed in our grid, specially in less densely populated areas. Examples are displayed in the figures below:

We have added this comparison of population estimations in the Supplemental Material (title: Comparison of estimated population with that from the WorldPop dataset, Fig S12) and in the Method section (Population on the grid subsection lines 326-330)

- 1.9. **Data availability:** As I understand the statement of data availability, it would be impossible to reproduce the work presented here. I would encourage the authors to make the dengue surveillance data available at sufficient spatial scale to reproduce some of the analyses in the paper. These are aggregated and at least 6 years old and so their sensitivity appears to be low. Perhaps the authors can arrive at some level of

aggregation to sufficiently deidentify data to allay privacy concerns but also make some analyses reproducible.

We understand the data availability concern and have now provided data to make our results reproducible. Specifically, we have provided datasets for: population density at the unit level, the time series of cases for units aggregated according to their population density, for the three levels of the administrative regions. We have also provided the time series of presence and absence of infection in each unit. The previous availability statement followed from what is permitted by health authorities of the city of Rio. We therefore indicated that those interested in the raw data should directly contact the city, per their conditions. However, the data we provide now allows for the complete reproduction of our results.

2. **Reviewer #2 (Remarks to the Author):** Romeo-Aznar and colleagues present an intriguing hypothesis to explain the relative sizes of successive peaks in dengue epidemics within subunits of a large city. By introducing a metapopulation modeling approach that aggregates by population density rather than spatial proximity, the authors present evidence for density-dependent susceptible depletion and importation rates interacting with seasonally-forced transmission at an intra-city scale.

I enjoyed reading this manuscript. There are some potential issues with the analysis that I think need to be addressed before the conclusions are supported. I will defer to other reviewers on dengue biology, for instance the assumption of neglecting heterotypic protection.

MAJOR COMMENTS

- 2.1. **Spark rates and Reporting rate.**

- 2.1.1. **As I understand it the analysis is done conditional on a reporting rate which is assumed to be constant across spatial units. The reporting parameter is not estimated from data but rather the analysis is run at a range of different fixed reporting rates. It would be good to be more explicit about this.**

This is correct. The reporting rate as well as the other parameters in the model are constant in space and with the exception of the spark rate, they are not estimated from these data. In the absence of evidence for spatial variation of the reporting rate, studies have typically considered it uniform over space, including large-scale studies (where units consist of cities to countries). The other referees have also raised questions on the reporting rate, an important parameter. Please see response to comment 1.3 on spatial variation, and to comment 1.2, more generally on the chosen range of values. Our main objectives are to document a novel empirical pattern of relevance to dengue epidemiology and possibly to other infectious diseases, and to provide an explanation for it, based on basic transmission models with “reasonable” parameters

values for DENV4 transmission in the city of Rio de Janeiro in 2012. It is indeed possible that there is a better combination of parameters that would capture this pattern even better, including for the reporting rate and the initial number of susceptibles. We have now provided better documentation and explanation for our parameter choices, and have considered via simulation the robustness to variation in these choices, with particular attention to the reporting rate. We have made more explicit that the parameters are not fitted to the data. They come from the literature on dengue epidemiology including earlier work of ours at an aggregated level for the city. Estimation of parameters for the metapopulation model indicated by this work is an on-going future direction.

(see changes in Method section lines 456-459, Discussion section lines 249-270)

2.1.2. How would it affect the analysis if reporting rate varied with population density? For example, what if reporting rate was systematically lower in denser areas of the city? Would that lead to higher estimates of the spark rate (σ_u) there? I would be curious to see the authors simulate this possibility to see if offered an alternative explanation for the data.

The possible dependence of the reporting rate on population density is an interesting issue concerning the robustness of our results. We do not view it however as an alternative explanation, in the sense that the explanation we have provided incorporates the basic elements of any seasonal SIR dynamics (seasonality of the transmission rate and the acquisition of immunity by the population), plus the observed timing of initiation of the epidemic. These simple components by themselves create the pattern in peak ratio (as the deterministic simulations show). A varying reporting rate would be additional to these basic components, and it could therefore affect the pattern but not create it on its own (depending on reporting rate range of variation and function form).

Because we lack evidence on the specific or even general form of this function (as well as on possible maximum and minimum values for the local reporting rates), we addressed this comment by assuming a decreasing linear relationship and by examining in detail access to reporting clinics, as described in the response to referee 1.

We run the stochastic model with a linear reporting rate that decreases with population as shown in Fig 6D below. Fig 6B shows that the resulting peak ratio pattern (and arrival time Fig 6A), for a reporting rate that varies between 0.3 and 0.5, is

similar to the one obtained with a constant reporting rate of 0.5. A better accuracy of this linear relationship than a constant reporting rate of 0.3 in denser units (compare also to Fig 4A and B of the main text) is because denser population quantiles are wide. That is, the denser population group includes population densities that go from about 3000 to 8000 people per unit and therefore a reporting rate that varies from about 0.44 to 0.3.

Fig 6. The purple color indicates results for the stochastic simulations with a reporting rate decreasing from 0.5 to 0.3 linearly with population density (B4), and the light blue color, those with a constant reporting rate of 0.5. B1) Mean arrival time as function of population (black circles correspond to values obtained from the data). B2) Peak ratio as a function of population density. B3) Representation of population quantiles (for 100 groups). B4) The linear relationship of the reporting rate with population density considered in the stochastic simulations.

We also considered whether there is evidence to assume that the reporting rate should vary significantly with population density for the city of Rio de Janeiro, where dengue transmission has been a major concern for public health authorities. We find that there are sufficient health units covering the city so that most people have access to one relatively close to their home, free of charge (see Fig 7A below). All these public clinics have to report dengue cases, as this is mandatory. Thus dengue notification includes all the people that seek assistance and does not result from a sample. That is, surveillance is passive. A bias could emerge though from people seeking assistance from private

clinics, a behavior that would apply to a small fraction of the population concentrated in the richer neighborhoods. Because rich neighborhoods in Rio can be very dense (e.g. Copacabana), and as much as poor communities (e.g. slums), we do not expect this sample bias to affect our results.

We specifically analyzed the access to healthcare clinics as a key factor affecting passive surveillance as implemented in Rio. We considered the distribution of the distances to the nearest public healthcare unit by population density group, shown in Fig 7B. As population density increases, there is a decreasing trend on the mean distance to the nearest public healthcare clinic, that goes from 1.6 km to 0.6 km. Despite the trend, the difference in mean distances is small. A maximum increase of 1km in distance should not impact the demand for medical assistance. We have highlighted the assumption of a uniform reporting rate in the Stochastic Model subsection (lines 456 to 459) and added this analysis (with more details than those presented here) in the Supplemental Material (title: Healthcare clinics' distance, FigS9-S11).

Fig 7. A) Map of the city showing the location of the public healthcare clinics (yellow dots) and the population density (population per unit). B) Box plot of the unit distance to the nearest public healthcare clinic as a function of unit population.

2.1.3. Estimates of spark rate involve counting the number of spatial units that do not have reported cases. I would think for a given prevalence p in a subpopulation of size N , the probability of getting exactly 0 reports during a unit of time is proportional to $(1-p)^N$, which suggests that spark rate estimates could vary with prevalence and population size. This sampling effect would be operating in addition to the differential impacts of demographic stochasticity in small populations. Can the authors reject the null hypothesis that the scaling relationships they describe are due to this effect?

*As the reviewer points out, the spark rate varies with prevalence and population. To address the possible “sampling effect” we explore our spark estimation for a toy simulation which explicitly considers the empirical prevalence. In this way, we can be sure that we are considering prevalence values consistent with the data and we can keep the assumptions simple (avoiding for example, any possible redundancy or hidden effect). We consider the “real” monthly cases in the unit u as: $C_u^{Sim}(t) = \text{Poisson}(p(t) * N_u)$, where $p(t) = C_{tot}(t)/(N_{tot} * \rho)$ is the prevalence in the city (where C_{tot} and N_{tot} and N_u values are taken from the data). Then we compute the observed cases as: $C_u^{Sim Obs}(t) = \text{binomial}(C_u^{Sim}(t), \rho)$. We produced one hundred simulations and then computed the number of sparks per unit as described in the Methods section of the main text. Fig 8 below shows that both the sparks estimated from “real” and “observed” cases have the same behavior with the total number of cases in the city (i.e. $C_{tot}^{Sim}(t) = \sum_u C_u^{Sim}(t)$ and $C_{tot}^{Sim Obs}(t) = \sum_u C_u^{Sim Obs}(t)$), either for a “large” ρ of 0.5 or a “small” ρ of 0.1. Since the good correspondence is obtained for all population groups, our procedure for the spark estimation should not produce a particular sampling effect acting on less dense units. (We note that because the real transmission may not be captured by a simple Poisson distribution of the cases weighted by population density, we obtain a somewhat different shape of the sparks and C_{tot} relationship than that in Fig 4C of the main text, in particular when C_{tot} is large. Also, we acknowledge that in this particular toy model, the reporting rate influences the “real” simulated cases but not the observed ones. We therefore have different combinations of prevalence and population size).*

A)

2.2. Writing / structure: The introduction develops the general importance of heterogeneity / structure in epidemic dynamics, but we get into the results without being clear what specifically the manuscript is going to test or how. Here are some suggested changes that may help

Thank you for these writing suggestions, which we have now adopted.

2.2.1. consider moving the first paragraph of results at the end of the introduction.

- 2.2.2. some phrases could be made less dramatic and more precise:
- 2.2.2.1. "daunting heterogeneity": but what is daunting about the heterogeneity has not yet been explained when this phrase is used.
 - 2.2.2.2. -"vast tracts of ground": is cinematic, yet has low information content: More precision would be better: by what measure are they large and larger than what?
 - 2.2.2.3. -similarly, "traditional coupling of spatially proximal localities on Euclidean grids"  "aggregating based on spatial proximity"

2.3. MINOR COMMENTS

- 2.3.1. please include line numbers and page numbers in the future
Done.
- 2.3.2. in main text: "ratio [of consecutive peak sizes] varies widely across the city" to Fig legend 1c: I would mention explicitly that the three panes show three different possibilities for peak ratios
We have now better indicated what the panels show, in the caption of the Figure 1. This clarification was indeed needed.
- 2.3.3. +WE HYPOTHESIZE THAT "two opposite variables shape the ratio of consecutive peaks"
Done.
- 2.3.4. population size is sometimes used when I think the authors always mean population density: eg when referring to figure 3. As size and density can have different impacts I think it is important to be consistent/precise, and consider defining and providing units for population density - is the denominator per 250x250m unit, or per square km or?
Thanks for the comment. We have now corrected this, and made explicit the units for the population density in both the text (for example line 103) and the figures axes.
- 2.3.5. Methods: Peak Ratio and Spatial Aggregation, should it be: "(10 areas, 33 regions and 160 neighborhoods)"?
We have changed this (line 342-343).
- 2.3.6. I don't see U- defined before it appears in eq 3. Is it the number of negative / uninfected units? Similarly, I don't see an explicit definition of what N_sparks is.
We have added these definitions in the development of the sparks rate equation (Stochastic Model subsection, see lines 411-413).

- 2.3.7. **around eqn 5, "to avoid statistical effects due to group size, we considered population quantiles." Which "statistical effects" are you referring to, and how does using quantiles avoid them?**

If groups are not of equal size, we can have statistically different effects. It is not the same, for example, to calculate a mean value on the basis of very different numbers of elements. Therefore, if we are comparing quantities such as mean values, aggregated time series, etc, it is prudent to have groups with the same number of elements. Since the population density distribution (number of people per unit) is not uniform, "equidistant" boundaries/breaks defining the groups result in sets with very different numbers of elements. In particular this effect becomes important when considering a large number of groups (for example for 100 groups, some of them have ~ 5 or fewer elements). Therefore, we decided to use population quantiles to define the population groups which, by definition, split a set into subsets with an equal number of elements. We have clarified this in the "Peak Ratio" and "Spatial Aggregation" subsection (lines 346-352).

- 2.3.8. **spark modeling: I found the methods through equations 2-4 hard to follow. In eqn 3 are you assuming something similar to a Levins metapopulation model in ecology where the number of new occupied patches is given by the recolonization rate multiplied by the fraction unoccupied patches? Similarly, "combining equations 2 and 3" to get 4 - smaller more explicit steps would be helpful**

We have added more steps (See Stochastic Model subsection). We hope this aspect of the methods is now clearer.

3. **Reviewer #3: In their paper Victoria Romeo-Aznar and collaborators are interested in the role of space in the propagation of a disease, the influence of both the heterogeneity in population distribution and the population movement. They offer a relevant and quite elegant approach to the complex topic the spatial dimension in epidemiology. For their demonstration they have used Dengue as an example and datasets from Rio de Janeiro. Specifically the authors analyzed the role of spatial scale of data aggregation in the propagation of a pathogen in relation to human density. With a new space aggregation based on population density they proposed that human density at fine spatial scales could explain the relative size of successive epidemic waves.**

I agree with the authors that transmission is first a local process and the local population structure is a key determinant of the pathogen propagation. And clearly this point is rarely adequately considered in the literature about Dengue. Thus one of the main advances of this manuscript is to highlight the importance of human density especially when human density is considered at a fine-scale. The demonstration by the authors concerning this point is very convincing for Dengue propagation in the Rio de Janeiro city.

Another very interesting novelty proposed by the authors is to describe the importation of cases similarly to the arrival of a "spark". This process appears

important to describe the transmission at a local level in absence of complete information about population movement. As stressed by the authors, this finding could suggest a novel formulation of metapopulation dynamics in complex urban environments.

The authors also found an invariant power law for the ratio of consecutive peak sizes when a serotype first enters in a given setting in relation to human density. This last point seems less convincing for several reasons.

We very much thank the referee for the positive evaluation of our work. For the last comment on the invariant power law, we wish to point out a potential confusion: (a) What is invariant in our findings is the nonlinear pattern for the dependence of the peak ratio on population density. That is, regardless of the number of groupings (population density quantiles) used to aggregate the city, as long as we aggregate according to population density, the pattern remains the same. This emphasizes the importance of considering the geography of the city in this way to explain and model the epidemiology. (b) The power law we demonstrate concerns the “spark rate” as a function of total cases in the city, and please note that the exponent of such power law does vary with population density. We do not claim that this pattern is invariant.

3.1. The authors also found an invariant power law for the ratio of consecutive peak sizes when a serotype first enters in a given setting in relation to human density. This last point seems less convincing for several reasons.

Please see our clarification in the general response above. The invariance is relative to the number of groupings and to the uncovered pattern of how peak ratio varies with population density. The power law is for the spark rate as a function of total cases in the city. Regardless, we appreciate the comments below and have now clarified these possible concerns.

3.1.1. First the hypothesis of a naïve population to a pathogen, DENV4 in this manuscript, that first enters in a given setting appears very restrictive. Indeed as most tropical and sub-tropical regions have submitted the four dengue strains the potential applications of these results appear very limited. This greatly limits the significance of the presented results. I think that authors must explore the possibility of relaxing this hypothesis for instance by considering re-emerging dengue strains with a low seroprevalence (10 or 20%).

This important issue was also raised by referee 1 in point 1.5. Please see the response to comment 1.5 above, where we provided evidence from the literature on the time of the first emergence of DENV4 in Rio de Janeiro and showed simulations with different initial numbers of immune individuals. We have now added these results to the manuscript. (see: Fig S14, new paragraph in the Discussion section lines 249-270, and added lines 180-186 in the Results section).

On the general observation that in endemic dengue regions the different serotypes co-circulate, we agree that it can limit the direct application of our work. We now acknowledge this issue in the Discussion and raise the question for future work. The additional results described above suggest however that the pattern we describe would still be relevant. Even if we were to consider our study as most directly applicable to seasonal epidemic dengue at the edge of its expanding geographic distribution, we believe the findings would still inform a considerable population and geographic area and one that is likely to grow with climate change. See also our response below and exploration of a data set for Delhi.

3.1.2. Secondly, there is just one example with two consecutive peaks... As DENV4 has emerged in different Brazilian cities, a second example, even with a coarser scale data, would reinforce this second main result.

We agree with the referee that exploring this pattern for other Brazilian cities would be really interesting. However, this kind of disaggregated dataset with very high spatial resolution remains rare and difficult to obtain. On the positive side, we expect our work to emphasize their importance and to stimulate surveillance efforts to record and share data at that level. Coarser-scale datasets, for example at the level of neighborhoods (which are also non trivial to obtain), would not allow one to detect the pattern and the influence of population density on successive waves, as we have shown in Fig 2B in our manuscript.

Although we were unable to find a similarly resolved dataset for another city of Brazil, we did have access to high resolution reported cases for the city of Delhi (India). By using the annual cases in the city of Delhi (at a 250m by 250m spatial resolution¹⁴), we computed the ratio of cumulative incidence in the seasons of 2009 and 2008. Because cases in this data set were reported annually, we considered the ratio of cumulative incidence rather than that of peak size, as referee #1 suggested (see comment 1.4 by referee 1, and our demonstration that the pattern we described holds when we measure the size of a given wave in this way). The calendar years correspond to the dengue transmission seasons in Delhi, which allows us to consider total yearly cases as a measure of epidemic size. The following figure (Fig 9) shows the behavior of this quantity with population

¹⁴ Telle, O., Vaguet, A., Yadav, N. K., Lefebvre, B., Daudé, E., Paul, R. E., ... & Nagpal, B. N. (2016). The spread of dengue in an endemic urban milieu—the case of Delhi, India. *PloS one*, 11(1), e0146539.

density, in a pattern that corroborates that of Rio de Janeiro. We note that Delhi represents a seasonally endemic location of the kind the referee mentions in 3.1.1. It is also a location with a less established surveillance effort than that of Rio de Janeiro. From that perspective, it is encouraging that we see a similar trend overall in the figure below, with the potential for it to be more noisy.

We have added this result in a new paragraph in the Discussion section (lines 272-278) and Supplemental Material (Fig S15).

- 3.1.3. **Thirdly, the use of the parameter “arrival time of infection” could be problematic because it is a parameter very difficult to estimate in real world, especially for a disease dominated by asymptomatic cases and in crowded settings of under-developing countries. The sensitivity to uncertainty associated with this parameter could be done.**

We fully agree with the referee and apologize for not being more clear on this. In fact, the sensitivity of the deterministic model to the value of the arrival time was one of our main motivations for considering simulations in a stochastic framework. To clarify this point further, we have added the following text in the article: “The deterministic nature of the model combined with the small size of the units makes simulations very sensitive to initial t_0 values. Small population sizes per se would introduce important demographic noise, here neglected, and the observed arrival times used in the simulations are likely delayed with respect to the first true local introduction of the virus. These limitations lead us to extend our analysis to a stochastic framework.”(lines 149-153). Importantly, in the stochastic model, the arrival time is not

fixed and specified, but implemented with the estimated “spark rate” as a Poisson process. This formulation accounts for the stochastic variability in the time of the introduction, while preserving the observed empirical trend in this time with population density as shown in Fig 4 (of the main text).

- 3.1.4. Fourthly, it is difficult to claim that the model found similar results to the observed ones when a key parameter of the model the “arrival time of infection” is obtained based on the same observations, observations that are used for the comparison with the model results (Fig 3B and page 6).**

The idea of showing the deterministic simulations first was not to claim the model success per se, but to exploit the simplicity of a deterministic SIR formulation to provide an intuition about the underlying processes that drive the observed peak ratio pattern. We thus used the model to illustrate the dynamical patterns that arise from the interplay of seasonality with arbitrary values of arrival times and population densities (Figure 3A, main manuscript). We then showed an empirical pattern in the arrival times as a function of population densities, which combined with the processes of immunity acquisition and seasonality, generates the peak ratio pattern. Finally, as we answered in the previous comment, we noted the limitations of this approach and moved to a stochastic model which by construction includes the arrival times through the estimated spark rate. The central argument of Figure 3 is that population density influences the interplay of seasonality and immunity acquisition in two opposite ways, which gives rise to the non-monotonic but clear dependence of successive wave size with population density. Our argument does not suffer from the suggested redundancy because what we are saying is that: given the dependence of arrival times on population density (captured as a stochastic process), SIR dynamics would naturally lead to different relative sizes of successive waves across space, when space is organized according to density.

- 3.1.5. Finally, the importance of the reporting rate: It seems that the best adequation between the stochastic model results and the observations is for a reporting rate equals to 0.5 (Fig. 4). However for a disease dominated by asymptomatic infections and in crowded and poor settings such as those we can find in Rio de Janeiro, the reporting rate should be very low... At least lower than 0.5...**

This is an important comment also raised by referee 1. Please, see response to comment 1.2 and new paragraph in the Discussion section, lines 249 to 270.

- 3.2. For my point of view the authors did not give enough weight to the importance of the seasonality, particularly in the Discussion. In the case of Dengue, seasonality has the crucial role of stopping the transmission and then the depletion of human susceptibles,**

We appreciate this request as we thought we had made seasonality one of the two major processes explaining the central pattern in the successive waves. We agree seasonality plays a crucial role in the ways the referee explains. Without it, there would be no pattern, and no role of population density in the sense studied here. We have now edited the manuscript and added a sentence in the Introduction to make clearer the central role of seasonality from the beginning (lines 97-100 and line 261).

- 3.3. Moreover the references about dengue spatial modeling quoted in the Introduction, could be completed by some others references, which seem to me important. Especially, Lourenço & Recker (2013) and Amaku et al (2016) that offer solution in a spirit quite similar with those of the authors. It is also the case for the reference of the SIR model used for dengue with just with a recent auto-reference [25], on a topic with a very expansive range of literature. (References:**

- Lourenço J, Recker M (2013) Natural, Persistent Oscillations in a Spatial Multi-Strain Disease System with Application to Dengue. PLoS Comput Biol 9(10): e1003308.
- Amaku, M., Azevedo, F., Burattini, M. N., Coelho, G. E., Coutinho, F. A. B., Greenhalgh, D., ... & Massad, E. (2016). Magnitude and frequency variations of vector-borne infection outbreaks using the Ross–Macdonald model: explaining and predicting outbreaks of dengue fever. *Epidemiology & Infection*, 144(16), 3435-3450.)

We appreciate these contributions which demonstrate the importance of spatial structure to the understanding of dengue dynamics. In particular, we find interesting the analysis on the contributions of the individual drivers to dengue dynamical patterns by Lourenco & Recker. We agree that the references related to spatial modeling should have been broader and we have added the suggested references to the article.

To summarize, the findings of this work are of potential interest in the context of spatial epidemiology. Nevertheless due to the aforementioned limitations, this manuscript is not ready, in my opinion, for publication. Once the imperfections have been removed, this manuscript, which presents a new and potentially interesting approach, could be published in a journal with a large audience such as Nature Communications.

- 3.4. Other comments:**

3.4.1. Abstract: “Models that exploit this emergent simplicity should afford improved predictions of epidemic waves.” This point is not

demonstrated in this manuscript... Except the prediction of the size of the second peak that is very restrictive considering the potential complexity of an epidemic wave.

We have now re-written this sentence to say “improved predictions of the size of local successive epidemic waves”. We do think that despite the potential complexity of an epidemic wave, what we are targeting here is the overall size or peak size. In that respect, we are referring to the emergent “simplicity” of what is needed to explain successive epidemic size (for example, that our work suggests we would not need the explicit spatial coupling at high resolution). We seem to need population density at high resolution but can treat spatial coupling as global. This writing refers specifically to that simplicity. We hope the edited writing satisfies the referee, as we are not implying we can capture everything in predictions. We further note that although the size of the second peak (relative to the first one) may appear restrictive, we chose this quantity as reflecting something fundamental about the interplay of immunity acquisition and seasonality. Since those are key processes of these dengue epidemics, we sought a quantity whose explanation would indicate we have captured this essential interplay.

3.4.2. Page 5: Regarding the neglected heterotypic protection: Is it reasonable hypothesis? Reference? This is an important point of discussion in multi-strains Dengue models. In your framework, what would the effect of heterotypic protection be?

We have now discussed this question in relation to the new results varying the level of starting immunity in the population. Heterotypic protection arising from a previous season dominated by another serotype would reduce the initial fraction of the population susceptible (see Fig S13 and S14, and new paragraph in lines 249-270).

3.4.3. Page 11: Are the presented results sensitive to the non-urban classification too?

We were not clear on what the referee meant by the “non-urban” classification. In the Supplemental Material (Fig S1) we show that the peak ratio is not sensitive to the non-urban classification. We have included the full extent of the municipality of Rio de Janeiro, included administratively in the city. This geographical extent includes different levels of urbanization, also associated with different levels of population density.

3.4.4. Page 13: $t_u \geq t_0$: t_0 the “arrival time of infection”?

We changed the notation to make this clear.

3.4.5. Page 13: “we specified no explicit spatial coupling between units”: Reference?

We do not specify a reference here, as this is by the design of our model. That is, we use the importation rate of infections as an alternative to model spatial coupling. In this way this quantity is treated as a parameter, and we can model each unit independently. The purpose of this step is to show that units will exhibit the documented pattern in peak ratio with population density. We then proceed to analyze determinants of the importation rate itself, and in particular its dependence on local conditions (population density) and global ones (global cases).

3.4.6. Pages 13-15; Specify the difference between U and u, or perhaps it's just a font problem?

The small u is an index and the capital U is the total number of units in the city. We have clarified this notation.

3.4.7. Figure 1: A scale for population density must be added.

Since the goal of Fig 1B is to show the differences when groups are created according to geographical or population density aspects, we think that a color bar indicating the population density is confusing. However, we have a population density map in the supplementary material (Fig S6) where the scale is indicated. We have now referred the reader to this information in the caption of Fig 1 (main text).

3.4.8. Figure 3A: Is the same scale for the y-axis for the 12 graphs?

Yes it is, because the y-axis is normalized by the maximum value. We have clarified this in Fig 3 caption (main text).

3.4.9. Figure 3: It is important to specify that the arrival times used in simulations are those obtained from the data.

Thank you for this suggestion. We have now done so for the deterministic simulations (line 148 and caption of Fig 3).

REVIEWERS' COMMENTS

Reviewer #1 (Remarks to the Author):

The revised manuscript is much improved. I appreciate the authors careful consideration and response to each of my and the other reviewers' comments. Clearly, a lot of work went into the response document (a real pleasure to read) and the revisions made to the manuscript. I feel that the manuscript will be a strong contribution to the literature on infectious disease ecology and dengue specifically.

Reviewer #2 (Remarks to the Author):

I think the revision is excellent. The authors have done a great job responding in detail to the referee comments. I am very enthusiastic about this paper and keen to see it published. At the most basic level, the correspondence between the empirically observed pattern in Figure 2A and the model output in Fig 3A,C is really striking. The development of the stochastic model, with spark rate as a connecting force is also well done. The attendant notions of emergent simplicity and the identification/construction of functional spatial epidemiological units is inspiring.

Beyond small suggestions below, my only remaining comment is on the use of the term "scale-invariant." I'd defer to the authors on this one, but my question is: as I understand it the claim of scale invariance is based on observing a similar pattern in peak ratios as a function of population density (Fig 2a) whether you use 12, 25, 50 or 100 'bins' for population density. Is this enough to claim scale invariance? I might be tempted to be more conservative and say the shape of the relationship did not appear sensitive to the choice of the number of groups over an order of magnitude, or something like that. But perhaps that is too conservative. Again I'd leave it up to the authors to decide if they feel the bar has been met for evidence of mathematical / statistical scale invariance.

Minor Comments:

In Fig2 legend, should it be "aggregated according to their population DENSITY.?"

Line 46: "Megacities spreading spatially enclose..." I don't understand this phrase

Line 48: "largely unexplored" effects of fine scale city structure, but see e.g. Eubank et al. Nature 2004? <https://www.nature.com/articles/nature02541>

Line 286: the effect result can IN SOME SYSTEMS? Be described

Line 341 typo - "time series" not "times series"

Reviewer #3 (Remarks to the Author):

I read with a great pleasure the new version of the manuscript. The authors have done an impressive job improving same. As a reader, the current version of the manuscript is according to my expectations, and I recommend it for publication.

Response to Referees

(Referees' comments in bold; our responses in italic)

Reviewer #1 (Remarks to the Author):

The revised manuscript is much improved. I appreciate the authors careful consideration and response to each of my and the other reviewers' comments. Clearly, a lot of work went into the response document (a real pleasure to read) and the revisions made to the manuscript. I feel that the manuscript will be a strong contribution to the literature on infectious disease ecology and dengue specifically.

Thank you very much.

Reviewer #2 (Remarks to the Author):

I think the revision is excellent. The authors have done a great job responding in detail to the referee comments. I am very enthusiastic about this paper and keen to see it published. At the most basic level, the correspondence between the empirically observed pattern in Figure 2A and the model output in Fig 3A,C is really striking. The development of the stochastic model, with spark rate as a connecting force is also well done. The attendant notions of emergent simplicity and the identification/construction of functional spatial epidemiological units is inspiring.

Thank you!

Beyond small suggestions below, my only remaining comment is on the use of the term "scale-invariant." I'd defer to the authors on this one, but my question is: as I understand it the claim of scale invariance is based on observing a similar pattern in peak ratios as a function of population density (Fig 2a) whether you use 12, 25, 50 or 100 'bins' for population density. Is this enough to claim scale invariance? I might be tempted to be more conservative and say the shape of the relationship did not appear sensitive to the choice of the number of groups over an order of magnitude, or something like that. But perhaps that is too conservative. Again I'd leave it up to the authors to decide if they feel the bar has been met for evidence of mathematical / statistical scale invariance.

We see the point. It is true that technically the term "scale-invariance" may go too far when considered in connection to its use in physics and in the literature on power law patterns and invariance over at least two orders of magnitude. We resolved the issue by eliminating the use of "scale-invariant pattern" and editing one sentence in the abstract, another in the opening paragraph of the Results, and two others in the Discussion.

Minor Comments:

In Fig2 legend, should it be “aggregated according to their population DENSITY.”?

Yes, we have now edited this sentence.

Line 46: “Megacities spreading spatially enclose...” I don’t understand this phrase

*Thank you for letting us know. We have now simplified the writing to:
“Megacities continue to grow spatially in ways that encompass pronounced heterogeneity in population density and movement, yet the effects on disease spread of the resulting fine-scale structure remain largely unexplored, with some notable exceptions based on individual-based models (Eubank et al.) .”*

Line 48: “largely unexplored” effects of fine scale city structure, but see e.g. Eubank et al. Nature 2004? <https://www.nature.com/articles/nature02541>

Thank you for this relevant reference; we have now added it in the text, as you can see from the sentence above.

Line 286: the effect result can IN SOME SYSTEMS? Be described

Yes, this makes sense. We have added these words.

Line 341 typo - “time series” not “times series”

ok. Corrected.

Reviewer #3 (Remarks to the Author):

I read with a great pleasure the new version of the manuscript. The authors have done an impressive job improving same. As a reader, the current version of the manuscript is according to my expectations, and I recommend it for publication.

Very happy to hear.